# Distinct structural features of *Pseudomonas aeruginosa* ATP synthase revealed by cryo-electron microscopy

Meghna Sobti[1,2], Adam P. Gunn [3], Simon H. J. Brown [4], Lauren Zavan[3], Vesper M. Fraunfelter [5,6], Amanda L. Wolfe [5], Christopher A. McDevitt [3], P. Ryan Steed[5] & Alastair G. Stewart [1,2] ✉

$F_1F_o$ ATP synthase is the ubiquitous enzyme that synthesizes cellular ATP by coupling proton-motive force with rotational catalysis. Structural differences between prokaryotic and eukaryotic ATP synthases offer potential targets for antimicrobial development. Here, we present the 2.0–2.4 Å resolution cryo-electron microscopy structures of the ATP synthase from *Pseudomonas aeruginosa*, an opportunistic bacterial pathogen capable of causing serious infections in humans. Our structures identify two distinctive features of this species' enzyme: a distinct binding site for the inhibitory ε subunit, and a coordinated metal ion capping the cytoplasmic proton channel. Lower-resolution maps of the enzyme following incubation with MgATP showed conformational rearrangements of the ε subunit during activation. Visualization of bound water molecules in the periplasmic half-channel supports a Grotthuss proton-transfer mechanism. Focused classification of the $F_o$ motor resolves distinct ~11° sub-steps in the c-ring, corresponding to protonation and deprotonation events. Functional analyses show that modifications to either the ε subunit or the metal binding site influence ATP synthesis and hydrolysis. Mass spectrometry analyses suggests that the physiological metal within the complex is zinc. Collectively, these findings define structural features of *P. aeruginosa* ATP synthase that could serve as targets for antimicrobial therapeutics.

$F_1F_o$ ATP synthase is the ubiquitous enzyme that synthesizes adenosine triphosphate (ATP), the "energy currency" of the cell[1–3]. This synthesis is achieved by coupling the ion motive force, generated through oxidative-phosphorylation and photophosphorylation, with the physical rotation of the membrane-bound $F_o$ motor. Rotation drives the soluble $F_1$ motor, inducing conformational changes at its active sites that catalyze ATP synthesis.

ATP synthases are composed of multiple subunits that show varying stoichiometries with the simplest examples, found in bacteria such as *Escherichia coli* and *Pseudomonas aeruginosa*, containing eight different subunits: α, β, γ, δ, ε, a, b, and c[4,5]. The $F_1$ motor consists of the $\alpha_3\beta_3\gamma\delta\varepsilon$ subunits (subunit stoichiometry shown with subscript numbers), whereas the $F_o$ motor comprises the $ab_2c_{10}$ subunits (Fig. 1a)[6]. The $F_o$ motor includes a stationary component (the $ab_2$ subunits) and a

[1]Molecular, Structural and Computational Biology Division, The Victor Chang Cardiac Research Institute, Darlinghurst, NSW, Australia. [2]School of Clinical Medicine, Faculty of Medicine and Health, UNSW Sydney, Sydney, NSW, Australia. [3]Department of Microbiology and Immunology, The Peter Doherty Institute for Infection and Immunity, The University of Melbourne, Parkville, VIC, Australia. [4]School of Science, Molecular Horizons, and the Australian Research Council Centre for Cryo-electron Microscopy of Membrane Proteins, University of Wollongong, Wollongong, NSW, Australia. [5]Department of Chemistry and Biochemistry, University of North Carolina Asheville, Asheville, NC, USA. [6]Present address: Department of Molecular Genetics and Microbiology, Duke University, Durham, NC, USA. ✉e-mail: a.stewart@victorchang.edu.au

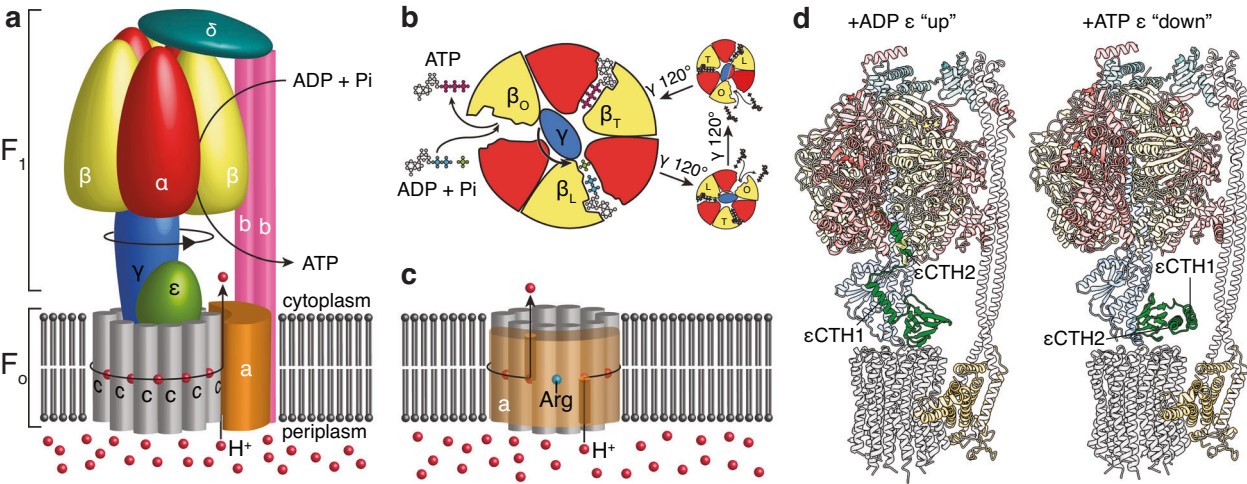

**Fig. 1 | Bacterial ATP synthase architecture and ε-mediated inhibition. a** A schematic representation of bacterial $F_1F_O$ ATP synthase in which subunits are labeled and color coded. Protons enter from the periplasm, driving the rotation of the c-ring. The central stalk (subunit γ, in blue) transfers this rotation to $F_1$, inducing conformational changes in its α and β subunits where catalysis occurs. **b** A schematic describing synthesis in the $F_1$-ATPase. A cross-section through the αβγ subunits shows that the catalytic subunits undergo conformational changes in response to γ subunit rotation. These changes facilitate the binding of ADP+Pi and synthesis of ATP. **c** A schematic describing proton driven rotation of the c-ring.

Protons enter a half-channel from the periplasm and bind sequentially to a carboxylate on a c subunit. The ring rotates, and protons exit via another half-channel open to the cytoplasm. Subunit a contains an arginine residue between the half-channels that prevents short-circuiting of the motor. **d** Observed structural changes in *E. coli* ATP synthase show the ε subunit transitioning from an extended "up" (PDB: 6OQW) to a condensed "down" (PDB: 8DBW) conformation upon addition of ATP. In the extended conformation of ε, the εCTH2 inserts into the $F_1$-ATPase head, preventing closure of a β subunit and potentially inhibiting rotation and thus catalysis.

rotary component (a ring of c subunits, known as the c-ring). Ions drive the rotation of the c-ring by entering a channel that is open only to the periplasm, binding to a conserved acidic residue on a c subunit[7], which then rotates anticlockwise (when viewed from the $F_1$-ATPase) until it aligns with a cytoplasmic half-channel, through which the ion is released (Fig. 1c)[8,9]. These two half-channels have been observed in ATP synthases from many species; the periplasmic channel is proposed to contain a chain of waters[10], whereas the cytoplasmic channel is more open[11], forming a large evagination in the membrane. The c-ring is joined to a central rotor (comprising subunits γ and ε) that transfers its rotation to the $F_1$ motor. In turn, the rotation of the central rotor induces conformational changes in the stationary $α_3β_3$ subunits of the $F_1$ motor, with three catalytic sites located at the interface between the α and β subunits, resulting in the conversion of adenosine diphosphate (ADP) and inorganic phosphate (Pi) to ATP[12–14] (Fig. 1b).

Regulation of ATP synthases is crucial, because an uninhibited enzyme can operate in reverse, wastefully hydrolyzing ATP and pumping protons[15]. ATP synthases from different organisms contain a spectrum of regulatory elements[16]. These range from auxiliary subunits that bind to the complex under certain conditions (e.g. IF₁ in mammals[17] and ζ in *Paracoccus denitrificans*)[18], internal subunits that may sense environmental changes and block the central rotor's rotation (e.g. subunit ε in *E. coli*)[19], to extensions of the catalytic subunits that can bind the central rotor and prevent rotation (e.g., *Mycobacterium tuberculosis*)[20,21]. In *E. coli*[19,22] and related organisms[23] the ε subunit has two domains that mediate distinct functions: (i) the N-terminal β sandwich[24] aids in coupling the $γεc_{10}$ central rotor; and (ii) the C-terminal helices, termed C-terminal helix 1 [εCTH1] and 2 [εCTH2], may regulate the enzyme[16]. In an active enzyme, the *E. coli* ε subunit adopts a condensed "down" state[24,25], allowing rotation and ATP synthesis. However, in an inhibited enzyme, transition[26] of the εCTH's to an extended "up" state[22,27] likely prevents its rotation in the $F_1$ motor (Fig. 1d).

Although some bacteria such as *E. coli* can bypass ATP synthase during ATP synthesis using fermentation, pseudomonads largely rely on ATP synthase even during fermentative processes[28], making the ATP synthase a potential target of antimicrobial therapies[5].

*P. aeruginosa* is a Gram-negative, biofilm-forming bacterium prevalent in hospital settings and particularly dangerous for patients with chronic lung disease, such as cystic fibrosis, or those with weakened immune systems[29]. Multidrug-resistant (MDR) *P. aeruginosa* is designated as a "serious threat" by the Centers for Disease Control in their 2019 report[30]. In the United States, *P. aeruginosa* infections are responsible for approximately 2700 deaths annually, with an estimated economic burden of US$767 million. The World Health Organization has designated *P. aeruginosa* as a high priority for new antimicrobial therapeutics[31]. Accordingly, the development of antimicrobials that specifically target its ATP synthase represents an attractive approach. However, current strategies have relied on studies of enzymes from related species in the absence of a high-resolution or mechanistic understanding of the pseudomonad enzyme, potentially limiting the accuracy of designed inhibitors.

Here, we present the structure of *P. aeruginosa* ATP synthase in multiple rotary conformations using cryogenic-Electron Microscopy (cryo-EM). The structures identify two distinct features of this species' enzyme and provide a detailed description of the catalytic site and proton pathway. These structural insights provide important background knowledge to facilitate the development of therapeutics targeting this crucial *P. aeruginosa* enzyme.

## Results
### Cryo-EM of *P. aeruginosa* ATP synthase
*P. aeruginosa* ATP synthase was expressed and purified from *E. coli* DK8 cells[32]; shown to be active and Lauryldimethylamine oxide (LDAO) sensitive in ATP regeneration assays (Fig. S1; rate = 7.6 ATP/$F_1F_O$/sec before and rate = 128.4 ATP/$F_1F_O$/sec after addition of LDAO); and examined by cryo-EM. Initially, to increase the likelihood of imaging an inhibited enzyme and thereby aid data processing, 10 mM MgADP was added prior to grid freezing, as performed in analogous studies[33]. Cryo-EM analysis yielded three distinct rotary states of *P. aeruginosa* ATP synthase, with the highest resolution map reaching ~2.0 Å (Figs. S2 and S3). Consistent with previous studies on related enzymes[33], these rotary states represent the catalytic $F_1$-motor in three rotary states. The cryo-EM maps were dominated by the larger $F_1$-

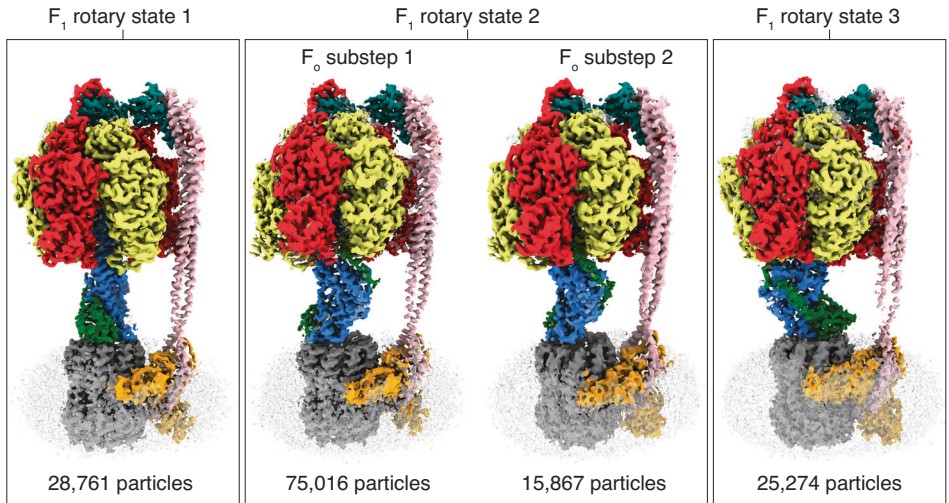

**Fig. 2 | *P. aeruginosa* ATP synthase in four distinct states.** Cryo-EM maps of *P. aeruginosa* ATP synthase, with subunits colored as in Fig. 1. The detergent micelle, in which complexes were purified, is shown transparent. These data identified four distinct states: three primary states corresponding to three rotational positions of the F$_1$-ATPase, each separated by 120°, and one state showing two positions of the F$_o$-motor with a ~11° rotation of the c-ring. Maps shown at 5 σ, and the number of particles in each class shown below.

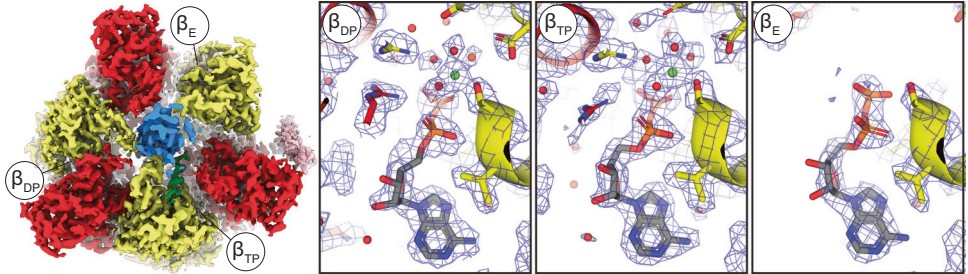

**Fig. 3 | Nucleotide binding in the F$_1$-ATPase.** Cryo-EM maps of the F$_1$-ATPase contain sufficiently high detail to identify nucleotides and associated water molecules. The F$_1$-ATPase map from State 1 (viewed from the membrane) and close-up views of the β subunit nucleotide binding sites (maps shown at 7.5 σ). β subunits labeled as β$_E$, β$_{DP}$, and β$_{TP}$, as in Abrahams et al. 1994[13].

motor, whereas the F$_o$-motor showed weaker density, likely due to flexibility and relative motion between the F$_1$ and F$_o$ motors. To improve the resolution of the F$_o$ motor region, a focused (masked on the F$_o$ region) 3D classification was performed, that improved the overall detail of the maps and allowed one of the maps to be subdivided into two F$_o$ rotary positions (Fig. 2, S2 and S3). Focused refinement (masked on the F$_o$ region) was performed on each rotary state, yielding maps with 2.4−3.0 Å resolution (Figs. S2, S3, and S4). These focused F$_o$ maps provide a highly detailed view of this region, which is explored in a later section. Previous studies have highlighted that rotary ATPases (which F$_1$F$_o$ ATP synthases are a subclass) are dynamic, with the catalytic "$_1$" motor tilting and rocking relative to the "$_o$" motor during rotation to facilitate smooth coupling[34–36]. Superposition of the *P. aeruginosa* ATP synthase rotary sub-states resolved here on the F$_o$ motor reveals that similar tilting and rocking of the F$_1$-ATPase occurs in this organism (Fig. S5), with these movements facilitated by twisting and bending of the peripheral stalk.

The 2 Å resolution map of *P. aeruginosa* ATP synthase allowed detailed model building of the F$_1$-motor region, highlighting water molecules around the nucleotide binding sites (Fig. 3). All three catalytic sites contained ADP, reflecting either the high concentration of ADP added prior to grid freezing or nucleotide retention during purification. The α, β, and γ subunits adopted positions nearly identical to those seen in bovine mitochondrial F$_1$-ATPase (PDB: 2JDI)[37], with an RMSD of 1.02 Å for the α, β, and γ subunits. Despite differences in nucleotide occupancy, the observed water network was very similar[37].

## ε subunit-mediated regulation of *P. aeruginosa* ATP synthase

Although the cryo-EM maps did not provide the highest detail for the central rotor, they were sufficient for model building and residue assignment. In the related *E. coli* enzyme imaged under similar conditions with the addition of 10 mM MgADP, the ε subunit's C-terminal helices are extended (Fig. 1d), with εCTH2 binding to the rotor and preventing the closure of β$_{DP}$ and rotation of γ[33]. Interestingly, in our cryo-EM maps of *P. aeruginosa* ATP synthase, we observed the εCTH2 in an "up" extended position but binding to the rotor from the opposite side of the γ subunit than that seen in *E. coli* and other organisms (i.e., 180° out of phase compared to other organisms: Fig. 4 and S6). In *P. aeruginosa* it binds into a gap formed by the α and β subunits and likely prevents rotation and catalysis acting as an autoinhibition mechanism. The binding pocket of the εCTH2 comprises the C-terminal domains of the α and β subunits and the foot of the γ subunit. The start of the εCTH2 binds to a cleft of the γ subunit that is formed by residues γ96-γ132. None of the residues in the cleft are identical to the human ATP synthase γ subunit (Fig. 4d), and the cryo-EM structure (PDB: 8H9S)[38] shows the cleft is filled by γHis120. Consequently, this cleft may present an interesting target for inhibitor design. A small molecule inhibitor targeting this pocket has the potential to prevent ε-mediated inhibition in *P. aeruginosa* ATP synthase. However, such a compound may not completely prevent bacterial growth and may require combination with other agents.

To further investigate ε mediated inhibition in *P. aeruginosa* ATP synthase, we performed cryo-EM on the same purified sample

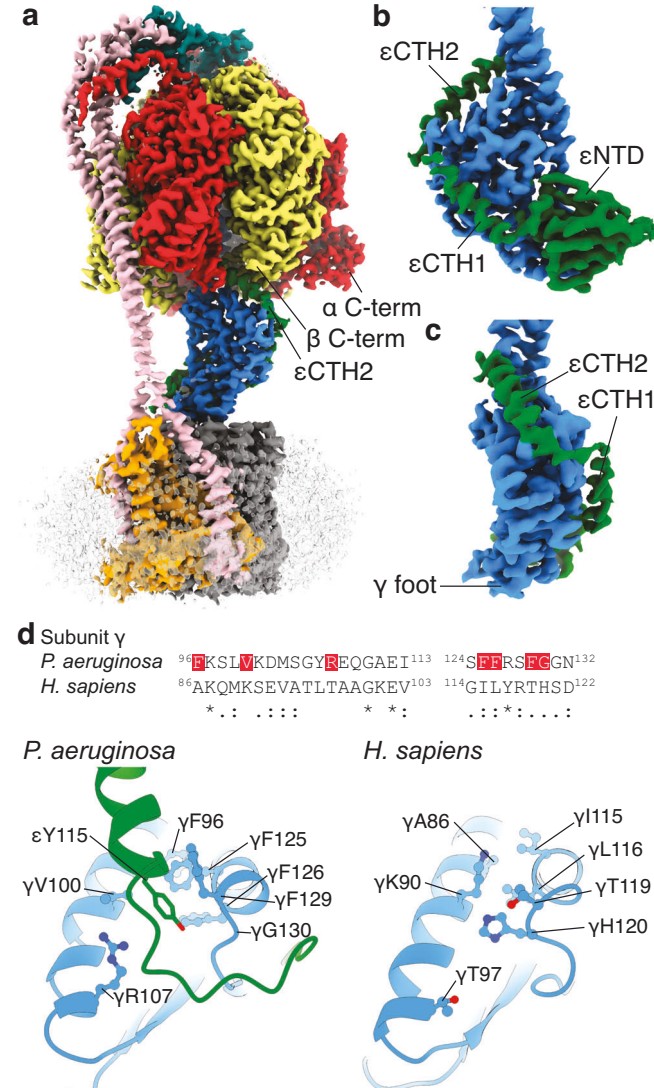

**Fig. 4 | The ε C-terminal domain binds to the rotor in a distinct extended conformation.** **a**–**c** Cryo-EM maps (F₁ rotary State 1 shown here) show the ε C-terminal domain in an extended conformation, tracing a path along the γ subunit to a position between two α and β subunits. This location has not been observed in other species and is on the opposite side of the complex compared to *E. coli* ATP synthase (Fig. S6). **a** Overall view of the complex, showing the location of εCTH2 and C-termini of an α and β subunit. **b** Extracted map of the ε and γ subunits, with the γ foot labelled. The εNTD forms a β sandwich fold and the εCTHs are in an extended configuration, wrapping around the γ subunit. **c** Rotated 90° relative to (**b**). **d** Sequence alignment of the subunit γ εCTH2 binding region of *P. aeruginosa* PA14 and *Homo sapiens* shows some conservation, although none of the residues in the cleft are identical. Structural comparison of this region reveals that the cleft is only formed in *P. aeruginosa*. A network of phenylalanines and other residues (highlighted red in the sequence alignment) creates a pocket for the εCTH2 region to bind in *P. aeruginosa*, whereas this cleft is closed in *H. sapiens*, with γHis120 filling the space.

following incubation with 10 mM MgATP instead of 10 mM MgADP (Figs. 5, S7 and S8). Although the resulting maps varied in resolution, State 2 yielded reconstructions at ~3.5 Å, sufficient to resolve the conformation of each subunit. Consistent with previous studies on the *E. coli* enzyme[25,26], we observed a transition of the εCTD from an extended "up" to a condensed "down" conformation, accompanied by a small rotation of the γ subunit. However, unlike the related *E. coli* enzyme (that was only seen in the εCTD "down" state after incubation with 10 mM MgATP), here we captured both

εCTD conformations in the same experiment, with the εCTD "up" structure near identical to that presented earlier in this study. Given the distinct location of the εCTD binding observed here, we explored the functional consequences of ε-mediated regulation by generating a construct lacking both C-terminal α-helices, termed the ε90Δ mutant (Fig. S9a). Subsequent functional studies using *E. coli* DK8 cells expressing the ε90Δ mutant showed slowed growth on succinate minimal medium (Fig. 6a), as well as inhibited ATP synthesis activity (Fig. 6b) and increased ATP hydrolysis (Fig. 6c) in inverted membrane vesicles. Hence, removal of the εCTD is detrimental to bacterial growth, possibly due to inhibition of activity and/or dysregulation.

## The F_o motor of *P. aeruginosa* ATP synthase

Overall, the molecular architecture of the *P. aeruginosa* F_o-motor was similar to that of other organisms[10,33,36,39–41], featuring a stationary subunit a and a rotating c-ring, although our maps do highlight a series of differences. The c-ring stoichiometry, which varies from eight[42] to seventeen[43] subunits across different kingdoms, was ten in the cryo-EM maps of *P. aeruginosa* (Fig. S10a). However, the C-termini of the c subunits differed from the other organisms observed to date and have an extension that traverses to an adjacent c subunit, capping the exterior of the c-ring at the periplasmic interface (Fig. S10b–d). The F_o-focused cryo-EM map provided a detailed description of proton transport, identifying a metal binding site, water molecules likely tracing the proton path of the periplasmic channel and sub-rotations of the c-ring (Figs. 7–9).

Previous studies on other species' ATP synthases have suggested a region in the periplasmic channel that may harbor a "water wire" that facilitates a Grotthuss mechanism[10], whereby protons are passed from one water molecule to the next. Our F_o-focused cryo-EM map identified waters in similar positions within the periplasmic channel, with three of these waters forming a chain from the periplasm to residue aAsp122 that corresponds to aAsp119 in *E. coli* where it is required for function. From here, a proton could be transferred to residue aHis262 (aHis245 in *E. coli*, that is also required for function), potentially protonating into the cavity adjacent to cAsp60 that also contains water, and where protonation of the carboxylate of cAsp60 would occur. Interestingly, the density corresponding to residue aHis262 was anisotropic, suggesting it may occupy two positions (Fig. S11): one closer to residue aAsp122 and the other further towards the water molecule.

Although the cytoplasmic half channel did not contain identifiable water molecules, the C-terminal tail of subunit a appeared to facilitate the coordination of a metal ion (Fig. 8b, c). This coordination capped the cytoplasmic channel of the F_o-motor, occupying the evagination observed in all other structures of rotary ATP synthases to date[11] (Fig. 7b and S12). Residues aGlu182, aHis186, aAsp287, and aHis289 coordinated around the ion presenting a tetrahedral-like coordination, with a spherical ion-like density seen at high threshold (Fig. 8c). The overall sequence of *P. aeruginosa* ATP synthase subunit a is very similar to that of *E. coli* and *Acinetobacter baumannii* ATP synthase subunit a (Fig. 8d). However, a single amino acid insertion (aAsn288) in *P. aeruginosa* ATP synthase appears to provide the flexibility to wrap around the ion and form a tetrahedral coordination of aGlu182, aHis186, aAsp287 and aHis289.

Using the *P. aeruginosa* PA14 *atpB* gene as a reference sequence, putative *atpB* genes were identified in a total of 1042 genomes in a database of 1082 *Pseudomonas* spp. with a high nucleotide and translated amino acid sequence conservation of 88.7% and 97.7% pairwise identity, respectively (Table S1 and Fig. S13a, b). The metal-ion coordination site of *atpB* is highly conserved across *Pseudomonas* spp. with three of the four coordinated residues strictly conserved in ~96% of all genomes (Table S1, Fig. S13c; red bars). In contrast, aAsp287 is only conserved in 72% of the *Pseudomonas* species. However, it is important to note that in the remaining 28% of genomes, aAsp287 is

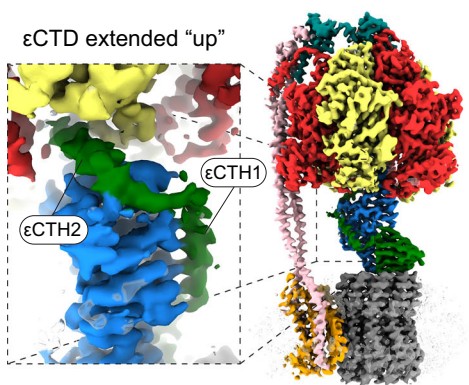
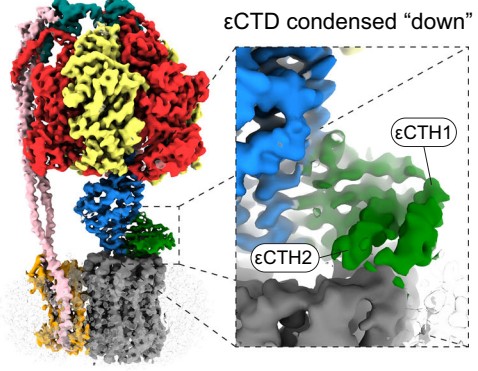

**Fig. 5 | MgATP induced ε-subunit rearrangements in *P. aeruginosa* ATP synthase.** Cryo-EM analysis of *P. aeruginosa* ATP synthase after incubation with 10 mM MgATP yielded multiple maps, capturing the enzyme in distinct rotary and inhibitory states (Fig. S7). Two maps of the enzyme in rotary State 2 are shown here, illustrating different conformations of the inhibitory ε subunit. Left: autoinhibited "up" extended state, preventing rotation. Right: active "down" condensed state, allowing the rotation of the γ subunit. Cryo-EM maps colored as in previous figures, with non-protein density transparent, and close-up views of the εCTD with helices labelled.

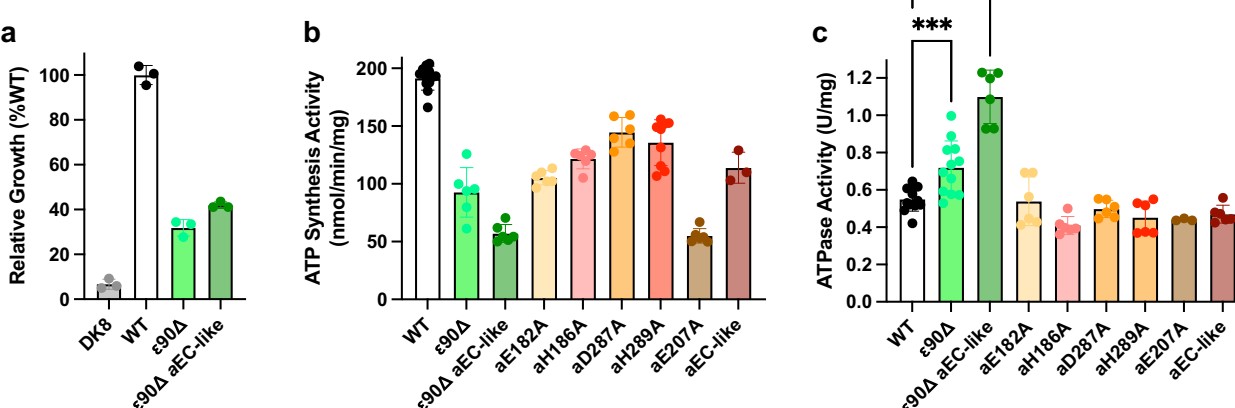

**Fig. 6 | Functional implications of εCTH2 and the cytoplasmic cap of *P. aeruginosa* ATP synthase. a** Transformants of *E. coli* DK8 grown on 0.6% succinate medium showed growth defects. ε90Δ (Subunit ε truncated after εLeu90 to remove the εCTD - Fig. S9a) and ε90Δ aEC-like (ε90Δ combined with mutations of a subunit to make it akin to the *E. coli* enzyme in sequence - Fig. 9b and Table S2) showed reduced growth compared to WT ($p < 0.0001$; one-way ANOVA) but greater than shown by untransformed DK8 cells. All $n = 3$. **b** ATP synthesis activity of inverted inner membrane vesicles showed reduced synthesis with all mutants compared to wild type ($p < 0.0001$; one-way ANOVA) (location of mutants shown in Fig. S9). WT; $n = 12$. ε90Δ; $n = 6$. ε90Δ aEC-like; $n = 6$. aE182A; $n = 6$. aH186A; $n = 6$. aD287A; $n = 6$. aH289A, $n = 9$. aE207A; $n = 6$. aEC-like; $n = 3$. **c** ATPase activity was largely unchanged for F$_o$ alone mutants. ε90Δ and ε90Δ aEC-like both show an increase in ATP activity versus wild type ($p = 0.0008$ and $p < 0.0001$ respectively; one-way ANOVA). WT; $n = 12$. ε90Δ; $n = 12$. ε90Δ aEC-like; $n = 6$. aE182A; $n = 6$. aH186A; $n = 6$. aD287A; $n = 6$. aH289A, $n = 9$. aE207A; $n = 6$. aEC-like; $n = 6$. Mean activities are plotted as bars, with individual activities plotted as points and ± SD. White for WT, greens for ε mutants and orange/reds for a mutants.

conservatively substituted by the carboxylate contributing aGlu287 residue in all but one genome, which is also frequently found at metal-coordination sites of this nature. Interestingly, variation in aAsp287 was only identified in non-*P. aeruginosa* strains within the analyzed database, while all *P. aeruginosa* strains contained aAsp287. Collectively, these data show that the residue comprising a metal-coordination site is present in almost all (>90%) *Pseudomonas* species analyzed (Table S1). To investigate the conserved nature of the ion-binding site beyond the pseudomonads, we also performed sequence analysis of the coordinating residues (aGlu182, aHis186, aAsp287, and aHis289) and aAsn288 insertion across a wide range of species (Table S2). These analyses showed that across the representative breadth of species analyzed, no other ATP synthase possessed a metal ion binding similar to that observed in *P. aeruginosa*.

Although it remains challenging to resolve the identity of ions based solely on cryo-EM maps[44], coordination chemistry and inductively coupled plasma-mass spectrometry (ICP-MS) can aid in identifying a putative ion. Hence, purified *P. aeruginosa* ATP synthase was analyzed by ICP-MS revealing molar excesses of $^{59}$Co and $^{66}$Zn (Fig. S14). Due to the purification of the complex using a cobalt-based resin, the presence of $^{59}$Co within the sample was anticipated. Overall, based on the amino acid composition of the metal-binding site, the preference of zinc ions for tetrahedral coordination, and the cellular abundance of first-row transition metals within *P. aeruginosa*[45], we inferred that the observed ion in this site was most likely zinc, although our methods cannot assign an identity to this ion unequivocally.

The precise function of this putative ion-binding site remains elusive, although the presence of this site at the proton channel suggests that it is likely to be involved in proton translocation. To explore this possibility further, we generated a series of mutants around the site that would weaken or remove metal binding. These were in the four side chains that directly coordinated the ion (aGlu182Ala,

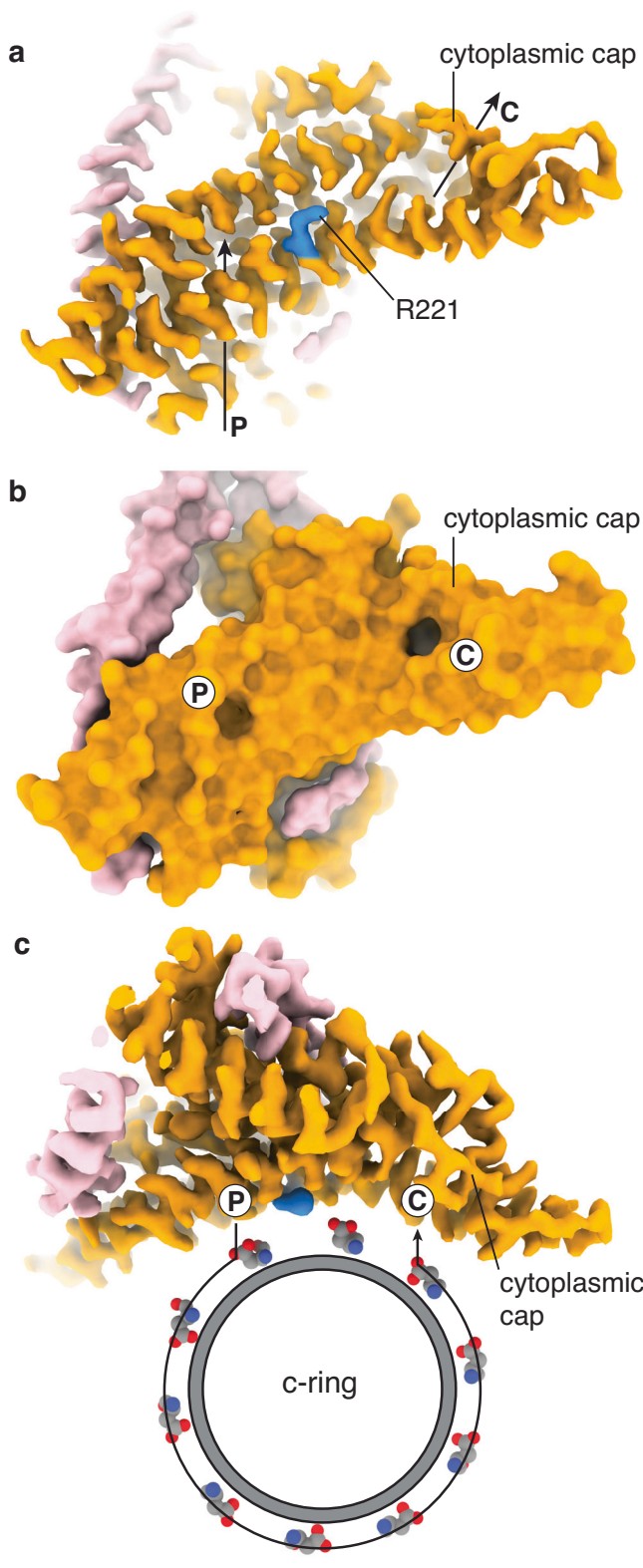

**Fig. 7 | Proton translocation pathway in *P. aeruginosa* ATP Synthase. a** The cryo-EM map reveals moderate resolution features within the $F_o$-motor, highlighting a periplasmic channel (P) and a cytoplasmic channel (C) that connect to the interior of the molecule. Residue Arg221 of subunit a (blue) is positioned between these two channels. The cytoplasmic channel is capped. Map shown at 8σ. **b** Surface representation of the molecular model illustrates two pores at the periplasmic protonation and cytoplasmic deprotonation sites. **c** In synthesis mode, protons from the periplasmic channel bind to residue cAsp60 of a c subunit (grey/CPK spheres) and rotate approximately 288° (8/10 c-ring rotation) before reaching the cytoplasmic channel. Residue aArg221 of subunit a prevents short-circuiting, ensuring the motor rotates anticlockwise (when viewed from the $F_1$-ATPase). Map shown at 8σ and rotated relative to (**a**, **b**), as viewed from the $F_1$-ATPase.

compared to WT (Fig. 6b), and the aGlu207Ala substitution decreased activity by >70%. None of the mutations substantially altered expression and assembly of $F_1F_o$ (Fig. S9), and all retained the ability to grow on non-fermentable minimal medium, although growth was slowed for aGlu207Ala (Fig. S9b). ATP hydrolysis assays on these mutants showed little to no effect, potentially suggesting that the coordination site is involved in synthesis rather than hydrolysis mode. However, as alanine substitutions can be relatively disruptive in their nature, the observed effects may reflect interference with proton translocation in this region rather than direct disruption of ion binding.

To our knowledge, this is the first identification of a metal ion within this region of an ATP synthase structure, including the closely related enzymes from *A. baumannii*[39] and *E. coli*[33]. This can be attributed to these related enzymes lacking the requisite combination of residues required for metal coordination, with each containing substitutions that would impair or abrogate metal-protein interaction at this site (Fig. 8d and Table S2). To determine the effect of an *E. coli*-like sequence in this region, which was not anticipated to support metal binding, we incorporated multiple mutations to alter the metal-binding loop to resemble subunit a of *E. coli*. Modifications included aHis186Gln and aAsp287Glu substitutions, and deletion of aAsn288. This mutant derivative construct, designated "aEC-like", showed ATP synthesis and hydrolysis activity similar to that of single alanine substitution mutants (Fig. 6), although this may have been due to lower expression levels (Fig. S16). To explore this mutant further and to see if it was linked to ATP hydrolysis, we also combined this aEC-like mutant with the εCTD removed mutant, designated "ε90Δ aEC-like". The mutant showed reduced growth on succinate medium, and the strongest reduction in ATP synthesis and the highest increase in ATP hydrolysis (Fig. 6). Together, these data suggest that both these regions may be involved in efficient synthesis and hydrolysis of ATP, and disrupting them results in dysregulation of the enzyme.

As well as defining the cytoplasmic and periplasmic half-channels, the cryo-EM maps also highlight a sub rotation of the $F_o$-motor. In State 2 (Fig. 2), focused classification methods produced two maps of the $F_o$-motor in which the c-ring had rotated relative to the a subunit (Fig. 9 and S4). Analysis of this rotation showed a rotation of ~11°, as calculated by superposing on the a subunit (Fig. S16). Notably, in one position, the proton-carrying carboxylate group on the c subunit (cAsp60) was located adjacent to the periplasmic channel, whereas in the other position, it is adjacent to the cytoplasmic channel (Fig. 9 and S14). This arrangement suggested that the enzyme pauses in each of these positions, potentially protonating at one and deprotonating at the other. Most interestingly, this rotation and proposed arrangement is near identical to that observed and hypothesized in single molecule rotation assays on *E. coli* $F_1F_o$ ATP synthase, where a proton translocation dependent 11° c-ring rotation steps have been observed[46]. The density for carboxylates in cryo-EM maps is often poor, and even though the resolution of these maps was relatively high, we could not assign unequivocally the rotamer position of the cAsp60 in all cases. However, the features of the maps suggested that the aspartate adopts different rotamers depending on its protonation

aHis186Ala, aAsp287Ala, and aHis289Ala) and a glutamate (aGlu207Ala) that appeared to orient the C-terminal aHis289 to facilitate binding (Figs. 8b, c and S9b). ATP synthase activity of WT *P. aeruginosa* ATP synthase in *E. coli* inverted membrane vesicles was measured to be $190 \pm 10$ nmol ATP/min/mg using a luciferase assay (Fig. 6b). Removal of any single coordinating side chain significantly decreased synthesis activity, causing 25–45% inhibition of activity

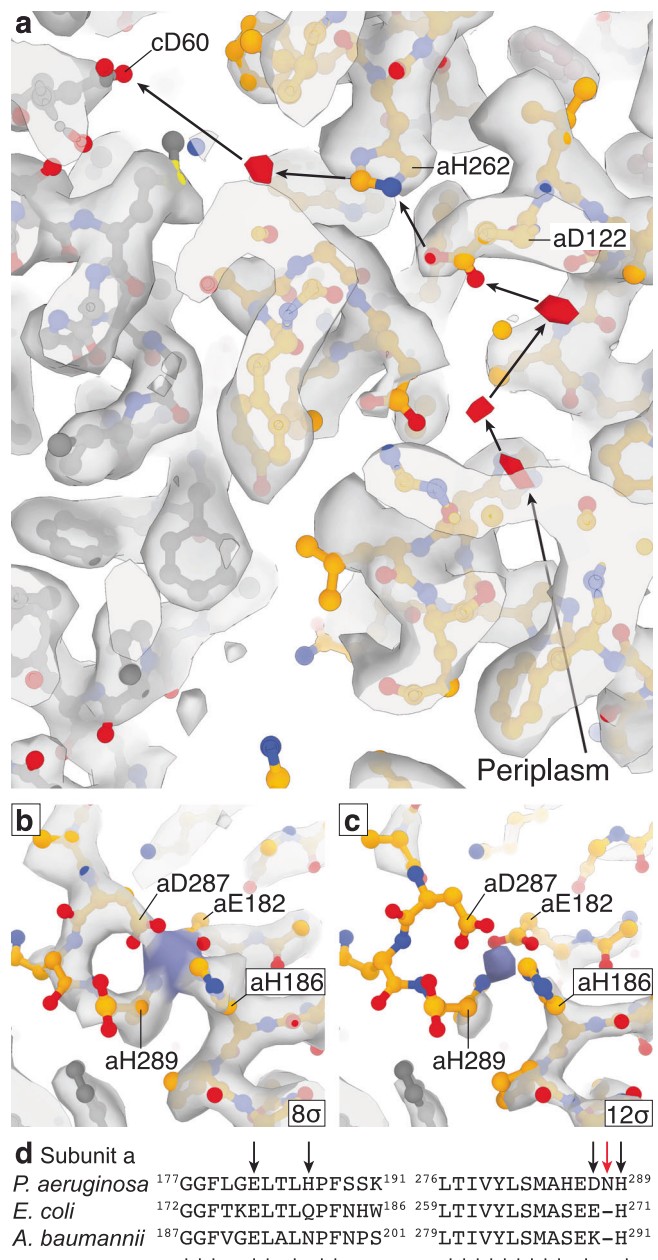

**Fig. 8 | The periplasmic channel and cytoplasmic cap. a** Density consistent with water molecules links the periplasm to cAsp60 via residues aAsp122 and aHis262. Map shown at 6.5σ, with possible water densities colored red. **b, c** The cytoplasmic proton channel is capped by the C-terminal tail of subunit a that wraps around a metal ion, forming a tetrahedral coordination. The map is shown at 8σ and 12σ to indicate the confidence of the map, with the possible metal ion density colored in purple. This feature has not been observed in other species (Fig. S12). **d** ATPB sequence alignments of *P. aeruginosa* PA14, *E. coli* K12 and *A. baumannii* ATCC 17978 show the lack of conservation in the metal coordination site residues aGlu182, aHis186, aAsp287, and aHis289. Metal coordinating residues are labelled with black arrows, and aAsn288, that allows aHis289 to participate in the metal ion coordination, is shown with a red arrow.

state and whether it is exposed to lipids or proteins, either pointing toward or away from the c-ring (Fig. S17), consistent with observations from crystallographic studies at varying pH[47].

## Discussion

In this study, we present detailed cryo-EM maps of *P. aeruginosa* ATP synthase, that show two distinctive features: a distinct site for ε-

mediated inhibition, and a metal-capped cytoplasmic proton channel. Additionally, these maps support a Grotthuss mechanism for proton movement through the periplasmic proton channel, and $F_o$ stepping with separated protonation and deprotonation events. Overall, the maps and models generated offer a template for understanding this organism and serve as a starting point for future antimicrobial development.

Interestingly, the binding mode of the *P. aeruginosa* ATP synthase ε subunit in the extended "up" position differed from that seen in related intact organisms (Fig. S6). However, a previous study[48] on isolated truncated rotor subunits of *E. coli* ATP synthase (a complex of subunit ε and residues 11–258 of subunit γ) showed a conformation reminiscent to that observed in *P. aeruginosa*. By contrast, structural studies on intact $F_1F_o$[27] and isolated $F_1$-ATPase[22] *E. coli* enzymes have all shown εCTD binding at the canonical site near where mammalian $IF_1$ binds[49]. This suggests that the observed conformation in the truncated subunits was likely due to the absence of other ATP synthase subunits. Despite extensive studies in related systems[25,50–52], the precise function of ε-subunit-mediated inhibition remains unclear. Nevertheless, our cryo-EM data showing a transition of the εCTD to a condensed "down" conformation following incubation with MgATP, combined with this functional data, does suggest that it is involved in regulating the enzyme. Given the distinct site of interaction between the εCTD and subunit γ, and that the εCTD truncation mutant showed growth inhibition, we propose that the εCTD binding pocket in *P. aeruginosa* is a potentially attractive target for future antimicrobial development against this species. However, it is important to note that this work measured the growth of *E. coli* cells expressing the *P. aeruginosa* enzyme from a plasmid. Therefore, we cannot be certain that disruption of ε regulation will mediate the same effect in the native organism.

In addition to being a distinct feature of *P. aeruginosa* ATP synthase that could be explored for specific drug binding, the capped cytoplasmic proton channel observed in this study raises questions about the physiological role of this region within the pseudomonad enzyme. Mutations in this region prevented synthesis from operating at peak efficiency (Fig. 6), and analyses of 1082 *Pseudomonas* species showed that it is highly conserved within this genus (Table S1) but absent from the ATP synthases of representative members of the *Gammaproteobacteri*. The metal coordination likely imposes greater order on the proton exit channel than observed in other ATP synthases[33,39]. Narrowing of this channel could facilitate proton movement through the cytoplasmic channel, directing protons to the shuttling residues (aLys214 and aGlu286) rather than disordered water molecules within a wider channel. aGlu207, which appears to hydrogen bond with aHis289 and support the metal coordination geometry, is the equivalent of aGlu196 in *E. coli*, which has been identified as sensitive to mutation[53], highlighting the importance of this region in proton transport. Recently, Dunkley and coworkers (2025)[54] described a His-rich C-terminal domain from subunit a in *Mesenchytraeus solifugus*, a glacial ice worm known to have high intracellular concentrations of ATP. When this His-rich extension was transferred to the *E. coli* enzyme it increased the rate of ATP synthesis. His-rich sequences are known to coordinate Zn ions, and here they may also provide a metal coordination site similar to that we observed in the *P. aeruginosa* enzyme, although the exact sequence is divergent in this region (Table S2).

In this study, the identity of the ion capping the cytoplasmic proton channel could not be determined unequivocally by ICP-MS or coordination chemistry. While the metal-binding coordinating residues would facilitate interaction with either cobalt and zinc, which were the two elements significantly enriched in the purified sample, the use of a cobalt-based resin for protein purification and the relatively low abundance of this element in *P. aeruginosa*[45], primarily as the metal chelate cobalamin, supports the inference that zinc is the most likely physiological metal cofactor. Despite this, it cannot be excluded

F$_o$ substep 1: cytoplasmic conducting

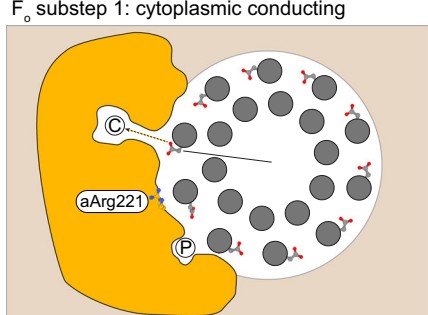

F$_o$ substep 2: periplasmic conducting

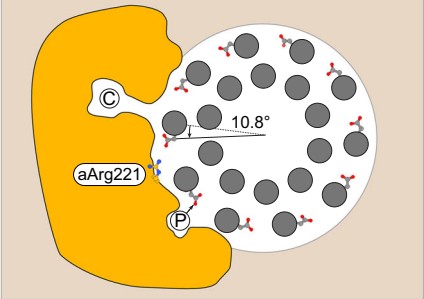

**Fig. 9 | F$_o$-motor conducting sub-states in *P. aeruginosa* ATP synthase.** Cryo-EM classification of F$_1$ rotary State 2 resolves two F$_o$ substeps (F$_o$ substep 1 and F$_o$ substep 2) that differ by a ~11° rotation of the c-ring (grey) relative to the stator subunit a (orange), as viewed from F$_1$. Left (F$_o$ substep 1): one cAsp60 (sticks) faces the cytoplasmic half-channel (C), enabling proton release (deprotonation). Right

(F$_o$ substep 2): the c-ring has rotated ~11° in the synthesis direction (black arrow), bringing a different cAsp60 into alignment with the periplasmic half-channel (P) for proton uptake (protonation). aArg221 is situated between the channels, preventing short circuiting of the motor.

that other cations could also interact with this site during periods of dysregulated metal ion homeostasis, which may influence ATP synthase function. In other cryo-EM and Molecular Dynamics studies on *Polytomella* ATP synthase[36,55], a Zn$^{2+}$ cation has been proposed to interact with aHis248 (aGlu230 in *P. aeruginosa*). This site is in the other half-channel (periplasmic) of the F$_o$-motor and appears to adopt slightly different positions as the c-ring rotates. However, in our cryo-EM study of the *P. aeruginosa* enzyme, we propose that the appearance of the map (Fig. 8a) is only consistent with waters in this region. Higher-resolution maps of ATP synthases from various species are now increasingly revealing metal ions bound within key protonation and deprotonation regions. This aspect of ATP synthase function likely requires further study, especially considering that some previous studies have used buffers containing chelating agents, which may have obscured metal binding.

In summary, our cryo-EM maps have provided a highly detailed view of ATP synthase from *P. aeruginosa*. We observe a distinct ε subunit inhibitory interface and a metal ion cap in the cytoplasmic channel. Both elements are absent in related enzymes, which may make these features potential targets for selective antimicrobial development. However, their effectiveness would require validation and compounds based on these regions may be ineffective without combination with other agents. Additionally, we have visualized directly a water-lined periplasmic half-channel consistent with a Grotthuss proton-transfer mechanism and captured discrete F$_o$ substeps that separate protonation and deprotonation. Together, these insights both advance our mechanistic understanding of ATP synthase and highlight potential new opportunities for antibiotic development.

## Methods
### Protein expression and purification
*P. aeruginosa* F$_1$F$_o$ ATP synthase protein was prepared by methods described in ref. 56. *P. aeruginosa* ATP synthase (plasmid details in ref. 57) was expressed in *E. coli* DK8 strain[58]. 15 L fermentation was run at 37 °C in LB medium supplemented with 100 µg/ml ampicillin for 5 h. The cells were harvested by centrifugation at 5000 g, providing ~90 g cells from 15 L culture, which were frozen for later use. ~15 g of cells was resuspended in lysis buffer containing 50 mM Tris/Cl pH 8.0, 100 mM NaCl, 5 mM MgCl$_2$, 0.1 mM EDTA, 2.5% glycerol and 1 µg/ml DNase I, and processed with three freeze-thaw cycles followed by one pass through a continuous flow cell disruptor at 20 kPSI. Cell debris was removed by centrifuging at 7700 g for 15 min, after which the membranes were collected by ultracentrifugation at 100,000 g for 1 h. The protein complex was extracted from membranes at 4 °C for 1 h by resuspending the pellet in extraction buffer

consisting of 20 mM Tris/Cl, pH 8.0, 300 mM NaCl, 2 mM MgCl$_2$, 100 mM sucrose, 20 mM imidazole, 10% glycerol, 4 mM digitonin, and EDTA-free protease inhibitor tablets (Roche). Insoluble material was removed by ultracentrifugation at 100,000 g for 30 min. The complex was then purified by binding on Talon resin (Clontech) and eluted in 150 mM imidazole and further purified with size exclusion chromatography on a 16/60 Superose 6 column equilibrated in a buffer containing 20 mM Tris/Cl pH 8.0, 100 mM NaCl, 1 mM digitonin and 2 mM MgCl$_2$. The purified protein was then concentrated to 11.7 µM (6 mg/ml), frozen, and stored for grid preparation. SDS PAGE was performed to assess protein purity (Fig. S18).

### Cryo-EM preparation and data collection of the 10 mM MgADP condition
Initial cryo-EM analysis of the complex was conducted at 200 kV, revealing similar but lower resolution features presented to this study. 1-(4-((((2-(benzylthio)quinolin-3-yl)methyl)amino)methyl)phenyl)-N,N dimethylmethanamine (compound #5 in Fraunfelter et al. 2023) was added in an attempt to visualize a complex inhibited by this compound. However, the compound was not observed in the resulting map. Despite this, the addition of compound #5 significantly improved the quality of the cryo-EM grids, facilitating data collection and ultimately enhancing the final map. Therefore, we present the 'apo' enzyme here in the presence of compound #5, as the compound was not detected in this study.

Compound #5 was dissolved in 100 % DMSO to give a stock solution of 30 mM. The compound was diluted 1:10 in water + 3 ul 10 mM MES (unadjusted pH) to give 2.3 mM working stock at pH 6.5. One microliter of 100 mM ADP/100 mM MgCl$_2$ (pH 8.0) and one microliter of the 2.3 mM compound #5 was added to an aliquot of 8 µl of purified *P. aeruginosa* F$_1$F$_o$ ATP synthase at 11.7 µM (Final compound concentration was 230 µM and final MgADP concentration 10 mM) and the sample incubated on ice for 1 hr, before 3.5 µl was placed on glow-discharged holey gold grid (UltrAufoils R1.2/1.3, 200 Mesh). Grids were blotted for 4 s at 22 °C, 100% humidity and flash-frozen in liquid ethane using a FEI Vitrobot Mark IV.

Grids were transferred to a Thermo Fisher Talos Arctica transmission electron microscope (TEM) operating at 200 kV and screened for ice thickness and particle density. Grids were subsequently transferred to a Thermo Fisher Titan Krios TEM G3i operating at 300 kV equipped with a Gatan BioQuantum energy filter (15 eV slit) and K3 Camera at Molecular Horizons, University of Wollongong. Images were recorded automatically using EPU at ×59,000 (displayed magnification of ×105,000 due to the energy filter), resulting in a pixel size of 0.83 Å. A total dose of 65 electrons per Å$^2$ was used spread over 80 frames, with a total exposure time of 6.41 s. 17,766 movie micrographs were collected.

## Cryo-EM data processing of the 10 mM MgADP condition

cryoSPARC[59] was used to perform all image processing and refinement. Micrographs were first motion corrected, and defocus was estimated using patches. After CTF estimation, poor quality exposures were discarded based on CTF fit, motion and ice thickness. Particles were automatically picked using a circular blob picker (min diameter 150, max diameter 200) and extracted with a box size of 400 pixels. Extracted particles were subjected to 2D classification to remove "junk" particles (such as particles within aggregates and minor contaminates). Ab initio 3D models were generated and particles further classified into 3 sub-states by subsequent rounds of heterogeneous refinements, reference-based motion correction and non-uniform refinements. Particles from each of the sub-states were further classified by 3D focused classification, with "force hard classification", using a mask encompassing the $F_O$ motor yielding another sub-state in State 2. The final maps were obtained by non-uniform refinements and local refinements using $F_1F_O$ or $F_O$ masks. FSC measurements were used to estimate the resolution of cryo-EM maps as a built-in feature of cryoSPARC[59]. See Figs. S2 and S3 for a summary of the data processing, local resolution, FSC curves and angular distribution plots and Table S3 for a summary of data collection and refinement statistics.

## Cryo-EM preparation and data collection of the 10 mM MgATP condition

For "with ATP" grid preparation, 0.5 μl of 100 mM ATP/100 mM MgCl$_2$ (pH 8.0) was added to an aliquot of 4.5 μl of purified *P. aeruginosa* F$_1$F$_O$ ATP synthase at 11.7 μM (Final protein concentration 10.53 μM and final MgATP concentration 10 mM) and the sample was incubated at 22 °C for 30 s, before 3.5 μl was placed on glow-discharged holey gold grid (UltrAuFoils R1.2/ 1.3, 200 Mesh). Grids were blotted for 4 s at 22 °C, 100% humidity, and flash-frozen in liquid ethane using a FEI Vitrobot Mark IV (total time for sample application, blotting, and freezing was ~45 s). Grids were transferred to a Thermo Fisher Talos Arctica transmission electron microscope (TEM) operating at 200 kV and screened for ice thickness and particle density. Grids were subsequently transferred to a Thermo Fisher Titan Krios TEM G3i operating at 300 kV equipped with a Gatan BioQuantum energy filter (15 eV slit) and K3 Camera at Molecular Horizons, University of Wollongong. Images were recorded automatically using EPU at ×59,000 (displayed magnification of ×105,000 due to the energy filter), resulting in a pixel size of 0.83 Å. A total dose of 74 electrons per Å$^2$ was used spread over 75 frames, with a total exposure time of 6.01 s. 33,804 movie micrographs were collected.

## Cryo-EM data processing of the 10 mM MgATP condition

cryoSPARC[59] was used to perform all image processing and refinement. Micrographs were first motion corrected, and defocus was estimated using patches. After CTF estimation, poor quality exposures were discarded based on CTF fit, motion and ice thickness. Particles were automatically picked using circular blob picker (min diameter 150, max diameter 200) and extracted with a box size of 400 pixels. Extracted particles were subjected to 2D classification to remove "junk" particles (such as particles within aggregates and minor contaminates). Ab initio 3D models were generated and particles further classified into three primary rotary states by subsequent rounds of heterogeneous refinements and non-uniform refinements. Particles from each of the three states were classified by 3D focused classification, with "force hard classification", using a mask encompassing the central stalk (subunits γ and ε) yielding the "up" and "down" conformations of subunit εCTD. The particles from "up" and "down" sub-states were further 3D classified using $F_O$ mask and "force hard classification". The final classes were refined using non-uniform refinements and local refinements with $F_1F_O$ focused masks. FSC measurements were used to estimate resolution of cryo-EM maps as a built-in feature of cryoSPARC[59]. See Figs. S7 and S8 for a summary of

the data processing, local resolution, FSC curves, and angular distribution plots and Table S4 for a summary of data collection and refinement statistics.

## Model building

Initial models were generated using Model Angelo[60] using a cryoS-PARC sharpened map and an input sequence for all the subunits. The models generated were manually adjusted and refined in Coot[61], PHENIX[62], and ISOLDE[63] (implemented in ChimeraX)[64]. Table S3 details the refinement and validation statistics.

## Calculation of relative c-ring rotation

Relative positions of the c-subunits were assigned based on their interaction with the N-terminal domain of subunit ε, under the assumption that this interaction remains constant during rotation. By superposing the models onto a subunit, we were able to observe the relative rotation of the c-ring (Fig. S16).

## Mutagenesis

Affinity-tagged and mutant ATP synthase complexes were constructed in the whole operon (*atpIBEFHAGDC*) *P. aeruginosa* ATP synthase expression plasmid pASH20[57] using synthetic gene fragments (Twist Bioscience). For mutations in subunit a (*atpB*), fragments including the desired point mutation(s) were inserted using the distinct HindIII, XhoI, and BamHI sites in and around *atpB*. Truncated subunit ε (*atpC*) was constructed by inserting an ochre stop codon after amino acid position 90 and ligating into pASH20 using the PpuMI and NdeI restriction sites. For affinity tag, an intermediate plasmid was constructed using megaprimer PCR[65] with a mutagenic primer to insert a translationally silent AflII restriction site in *atpD*. A gene fragment encoding a His6 tag on the N-terminus of subunit β (*atpD*) was then ligated into this intermediate between the PflMI and AflII sites. DNA sequences were verified by whole plasmid sequencing (Plasmidsaurus).

## Growth on succinate minimal medium

*E. coli* DK8 transformed with pASH20 (Fraunfelter, 2023) or its derivatives were streaked on LB agar with 100 μg/mL ampicillin. A single colony was used to inoculate 5 mL M63-TIV (61.8 mM KH$_2$PO$_4$, 38.2 mM K$_2$HPO$_4$, 15 mM (NH$_4$)$_2$SO$_4$, 1 mM MgSO$_4$, 1 μg/mL thiamine, 0.2 mM isoleucine, 0.2 mM valine) supplemented with 0.1% (w/v) glucose, 5% (v/v) LB, and 100 μg/mL ampicillin. After overnight growth at 37 °C with shaking, 10 μL was used to inoculate 250 μL M63-TIV supplemented with either 0.04% (w/v) glucose or 0.6% (w/v) succinate in a clear, sterile 96-well plate. The plate was incubated at 37 °C with shaking, with absorbance readings (optical density at 550 nm [OD550]) measured every 30 min in a BioTek Synergy H1 multimode plate reader. Bacterial growth at 6 h post inoculation was defined as the maximum and used to calculate growth relative to the wild type (DK8 pASH20). See Fig. S19 for relative growth comparison and Fig. S20 for growth curves.

## Preparation of inverted membrane vesicles

Inverted inner membrane vesicles were prepared from DK8 transformant strains as previously described in ref. 66. Protein concentrations were determined using the Lowry method.

## ATP synthesis activity

Measurement of ATP synthesis activity in inverted membrane vesicles was adapted from previously described luciferin/luciferase assays[57,67]. In a 96-well plate, ATP synthesis was initiated by adding 12.5 μg vesicles to a solution containing 5 mM Tricine-KOH, pH 8.0, 50 mM NaCl, 2.5 mM MgCl$_2$, 0.1 mM ADP, 3.75 mM K$_2$HPO$_4$, and 2.5 mM NADH (250 μL final volume). At 1, 2, 4, and 8 min, 50 μL was removed and mixed with 200 μL 12.5% trichloroacetic acid to quench the reaction. An equal volume of ATP standard solutions was also mixed with stop

solution and carried through subsequent steps. The stopped samples were diluted 100-fold with water and 10 μL was added to 100 μL 25 mM tricine-NaOH, pH 7.8, 5 mM MgSO$_4$, 0.1 mM EDTA, 0.1 mM NaN$_3$, 1 mM dithiothreitol, 150 μg/mL L-luciferin, and 7.5 μg/mL QuantiLum luciferase (Promega) in a white 96-well plate. Luminescence was measured after 2 minutes in a BioTek Synergy H1 multimode plate reader. In each assay, each sample was measured in triplicate and once more with 2 mM carbonyl cyanide m-chlorophenyl hydrazone (CCCP) to quantify the gradient-independent background concentration of ATP. For analysis, luminescence values were converted to nmol ATP using the ATP standard curve, and the slope of the linear range (typically 0-4 min) was used to calculate synthesis activity (in nmol ATP/min/mg) after subtracting the slope of the CCCP-containing sample over the same time range. Representative data in Fig. S21.

### ATP hydrolysis activity

**Inverted membranes**. ATPase activity of inverted membrane vesicles was determined using an ATP regeneration assay adapted from Milgrom and Duncan[50]. An assay buffer containing 20 mM Tris-acetate, 3.2 mM Mg(CH$_3$COO)$_2$, 10 mM KCH$_3$COO, 0.2 mM EDTA, 0.3 mM NADH, 1 mM PEP, 5 mM KCN, 7 μM FCCP, 0.1 mg/mL lactate dehydrogenase, 0.1 mg/mL pyruvate kinase, pH 8.0, was distributed in a 96-well plate and absorbance was measured at 340 nm and 420 nm in a BioTek Synergy H1 multimode plate reader at 30 °C. After 5 min, 1 μg of membrane vesicles were mixed into each well. After an additional 5 min, ATP was mixed into a final concentration of 1 mM, and absorbances were monitored for an additional 15 min. ATPase activity in U/mg, where 1 U is 1 μmol ATP/min, was calculated from the linear portion of the ΔA(340 nm–420 nm) trace after ATP addition using a molar absorptivity of 6317 M$^{-1}$ cm$^{-1}$ for NADH and a 0.75 cm pathlength (300 μL volume). Statistical significance of activity differences between WT and mutants was determined in GraphPad Prism using a one-way ANOVA.

**Purified protein**. ATP regeneration assays were performed as in Sobti et al.[56]. Reaction mix containing 100 mM KCl, 50 mM MOPS pH 7.4, 1 mM MgCl$_2$, 1 mM ATP, 2 mM PEP, 2.5 units/ml pyruvate kinase, 2.5 units/ml lactate dehydrogenase and 0.2 mM NADH was monitored for OD at 340 nm, 37 °C for 1 min. 8.4 μg of detergent-solubilized protein was then added and OD recorded for 5 min. 0.4% final concentration LDAO was then added to reaction mix and change in rate of activity was monitored and analysed (Fig S1).

### Western blotting

Samples of inverted membrane vesicles were mixed with an equal volume of 2x Laemmli running buffer (Bio-Rad), and 10 μg was run on an AnyKD precast SDS-PAGE gel (Bio-Rad) with prestained markers. Protein was wet transferred to a PVDF membrane at 75 V for 90 min in Towbin buffer (25 mM Tris, 192 mM glycine, 20% (v/v) methanol). The membrane was blocked with 5% dry milk (w/v) in TBST (20 mM Tris, 150 mM NaCl, 0.1 % (v/v) Tween-20), immunostained with 1:4000 rabbit anti-subunit β (Agrisera) in TBST containing 2% (w/v) BSA, and immunostained with 1:8000 AP-conjugated donkey anti-rabbit IgG in TBST containing 2% (w/v) BSA. Bands were developed using BCIP/NBT one-component solution (Surmodics).

### Inductively Coupled Plasma-Mass Spectrometry (ICP-MS)

Samples of ATPase (50 μL, containing ~350 μg protein) were prepared for elemental analysis by desalting in size exclusion chromatography buffer with Zeba 7 K MWCO spin columns (Cat. #89877, Thermo Scientific) and then dried in a vacuum centrifuge (Savant Speedvac, Thermo Scientific). Dried samples were acid hydrolyzed by heating at 95 °C in suprapure nitric acid (HNO$_3$, Cat.#100441, Merck) for 15 min in a dry block. Samples were diluted 20-fold with ultra-high purity water (MilliQ system, >18.2 Ohm) and heated for a further 15 min at 95 °C, then cooled on ice for 10 min. Samples were centrifuged for 5 min at

5000 g and the supernatants (top 950 μL) were transferred to a fresh tube (Technoplas 1.5 mL PP that had been rinsed with 1% (v/v) HNO$_3$) for immediate analysis. Elemental measurements were performed using an Agilent 8900 inductively coupled plasma-mass spectrometer (ICP-MS; Agilent Technologies) as previously described in Neville et al.[68] and Ganio, et al.[69]. The instrument was calibrated for elements of interest using mixed calibrator solutions (Multi-element Calibration Standard 2 A and 4, Agilent Technologies) diluted in 3.25% (v/v) suprapure HNO$_3$, acquiring signals for $^{24}$Mg, $^{44}$Ca, $^{55}$Mn, $^{56}$Fe, $^{59}$Co, $^{60}$Ni, $^{63}$Cu, $^{66}$Zn, $^{78}$Se, $^{95}$Mo, and $^{111}$Cd. Yttrium ($^{89}$Y; 100 μg L$^{-1}$ in 1% v/v HNO$_3$) was introduced as the internal standard via a T-piece positioned after the peristaltic pump. The instrument was tuned using commercially sourced tuning solution (PN 5185-5959, Agilent Technologies) according to the manufacturer's instructions. All element signals were acquired in He collision mode (4.3 mL min$^{-1}$), except for Fe and Se, which were acquired in H$_2$ mode (4.0 mL min$^{-1}$). Raw data analysis was performed using MassHunter v5.3 (Agilent Technologies), exported for stoichiometry calculations in Microsoft Excel (v2502, build 18526.20168), and GraphPad Prism 10.

### Sequence analysis

Sequencing for subunits were retrieved from UniProt[70] and aligned using ClustalW[71].

### Bioinformatic analysis

To assess conservation of the metal-binding coordinating residues in subunit a, a database of 1082 publicly available *Pseudomonas* spp. genomes was screened using the BLASTN screening tool, Screen Assembly (v1.2.7)[72], applying cut-offs of 80% identity and 80% reference length. Nucleotide and translated protein sequences were used for variation analysis by Geneious and MUSCLE Alignment (v5.1)[73] respectively in Geneious Prime with default settings. Aligned nucleotide sequences were trimmed using Noisy[74] (v1.5.12.1), and a phylogenetic tree was constructed using the maximum likelihood approach with IQ-Tree incorporating 1,000 ultrafast bootstrap[75] (v3.0.1). Trees were visualized using iTOL[76] (v7.2.1). Conservation of translated protein sequences was mapped to the tertiary structure of AtpB using UCSF Chimera[64] (v1.19). Amino acid frequency was graphed using GraphPad Prism (v10.4.1).

### Statistical analysis

FSC measurements were used to estimate resolution of cryo-EM maps as a built-in feature of cryoSPARC[59]. Details of this analysis can be found throughout the results and supplemental information. The program applied the "gold standard approach", where FSC is calculated from independently refined half reconstructions. Statistically significant differences between ATP synthesis activities of each mutant (3<n < 7) vs. wildtype (n = 12) were determined in GraphPad Prism using a one-way ANOVA assuming equal Gaussian distributions.

### Reporting summary

Further information on research design is available in the Nature Portfolio Reporting Summary linked to this article.

## Data availability

The *P. aeruginosa* ATP synthase +MgADP models and maps have been deposited under the Protein Data Bank (PDB) codes 9O19, 9O1A, 9O1B, 9O1C, 9O1D, 9O1E, 9O1F, 9O1G, 9O1H, 9O1J, 9O1K and Electron Microscopy Data Bank (EMDB) codes 49999, 70000, 70001, 70002, 70003, 70004, 70005, 70006, 70007, 70009, 70010 respectively. The *P. aeruginosa* ATP synthase +MgATP maps have been deposited under EMDB codes 71967 and 71968. Uncropped images of the SDS PAGE and Western blot in figs. S18 and S15 are provided in the source data. An Excel spreadsheet containing all raw data for Figs. 6, S1, S14, S19, S20 & S21 is provided in the source data, along with *p*-value calculations for Fig. 6.

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

## Acknowledgements

We wish to thank and acknowledge the use of the University of Wollongong Cryogenic Electron Microscopy Facility at Molecular Horizons, as well as the use of the Victor Chang Cardiac Research Institute Innovation Center (funded by the NSW Government) and the Electron Microscope Unit at UNSW Sydney. We also wish to thank and acknowledge Ian Reininger, who performed preliminary experiments on this project. Molecular graphics and analyses performed with UCSF ChimeraX, developed by the Resource for Biocomputing, Visualization, and Informatics at the University of California, San Francisco, with support from National Institutes of Health R01-GM129325 and the Office of Cyber Infrastructure and Computational Biology, National Institute of Allergy and Infectious Diseases. A.G.S. and C.A.M. are supported by National Health and Medical Research Council grants, 2016308 and 2036811, respectively. This research was partially funded by the Australian Research Council through Discovery Project DP250101405. This research was conducted by the Australian Research Council Industrial Transformation Training Centre for Cryo-Electron Microscopy of Membrane Proteins for Drug Discovery (IC200100052). Research was supported by National Institutes of Health grants R15 AI163474 to A.L.W. and R15 GM134453 to P.R.S.

## Author contributions

M.S. purified the protein and performed the cryo-EM study. A.P.G. and C.A.M. performed the ICP-MS study. S.H.J.B. aided in the collection and data processing of the cryo-EM data. L.Z. performed the sequence analysis. V.M.F. cloned the *P. aeruginosa* ATP synthase operon. P.R.S. performed the ATP synthesis and hydrolysis assays. A.L.W., C.A.M., P.R.S., and A.G.S. supervised the research. A.G.S. drafted the initial manuscript, and all authors edited the manuscript.

## Competing interests

The authors declare no competing interests.
