## [Transparent Peer Review file · Nature Communications]

Distinct Structural Features of *Pseudomonas aeruginosa* ATP Synthase Revealed by Cryo Electron Microscopy

Corresponding Author: Dr Alastair Stewart

Version 0:

Reviewer comments:

Reviewer #1

(Remarks to the Author)

Title: Distinct Structural Features of *Pseudomonas aeruginosa* ATP Synthase Revealed by Cryo Electron Microscopy

Journal: Nature Communications, NCOMMS-25-33756

All Authors: M. Sobti, A.P. Gunn, Simon, H.J. Brown, L. Zavan, V.M. Fraunfelder, A.L. Wolfe, C.A. McDevitt, P.R. Steed and A.G. Stewart

The *Pseudomonas aeruginosa* F1FO-ATP synthase is essential for the non-fermenting pathogen and therefore for the majority of the synthesis of Adenosine triphosphate (ATP). Recent studies by one of the teams have shown, that the c-ring of the FO-domain can be target by an inhibitor, resulting in ATP depletion of inverted membrane vesicles (IMVs; Fraunfelder et al., 2023). In the presented manuscript by Sobti et al, the authors describe the purification of the *P. aeruginosa* F1FO-ATP synthase, the determination of its atomic structure using cryo-EM, design, generation and characterization of the C-terminal truncated ϵ subunit mutant. The study is in line with significant contributions of the Stewart lab to the structural and mechanistic understanding of the *Escherichia coli* F-ATP synthase.

Comments:

- As shown by the Fraunfelder et al., 2023 publication (ACS Inf. Dis.), IMVs of *P. aeruginosa* show moderate ATP-driven proton-pumping activity, indicating ATP hydrolytic activity of the *P. aeruginosa* F1FO-ATP synthase. Therefore, an ATP hydrolysis experiment of the purified enzyme in the presence and absence of LDAO would add important information to this manuscript in regard to the activity of the heterologous produced enzyme as well as its ATPase latency and would allow a comparison to the F-ATP synthases from *Mycobacteria* and *Acinetobacter baumannii*.
- In this context, ATP-driven proton-pumping activity and ATP hydrolysis measurements of the wild-type (WT) and $\epsilon 90\Delta$ mutants IMVs would provide more information about the effect of the C-terminus to both activities and the argument that ATPase inhibition is mainly caused by this ϵ CTD.
- The deletion of the ϵ CTD dropped NADH-driven ATP synthesis of the IMVs by about 50%, which is nice. However, considering that the Krebs cycle will contribute to ATP formation, this mutant would still be able to produce about 60% within the pathogen under normal grow conditions, enabling the pathogen to replicate and survive. Figure S10 shows no major effect of this deletion in cell growth under Glucose conditions. Even if a compound would show optimal binding, it would not prevent the bacterium from growth and most importantly would not be bactericidal. This aspect is important for target validation and should be discussed.
- Similarly, the aGlu182Ala, aHis186Ala, aAsp287Ala, aHis289Ala and aGlu207Ala mutants showed some degree of ATP synthesis inhibition compared to the WT IMVs and no major drop in cell growth no matter whether succinate or glucose was present (accept aGlu207Ala under succinate conditions).
- In this context, it is written in figure legend 7B: "The cytoplasmic proton channel is capped by the C-terminal tail of subunit a, which wraps around a metal". This channel includes aGlu182Ala, aHis186Ala or aAsp287Ala. Could it be that the inhibition observed in ATP formation is in part due to the effect in proton transduction and not mainly because of the Zn²⁺-ion?
- In a cryo-EM structure of *Polytomella* sp. (Murphy et al. (2019) Science) a Zn²⁺-ion near residue aH248 was also resolved, which takes slightly different positions in cryo-EM states P1 and P2. Such data were used in MD simulation studies by Blan and Hummer (PNAS 121 (2024)). These results should be discussed in context to the Zn²⁺-binding and possible role found

for the *P. aeruginosa* F1FO-ATP synthase.

• Minor points:

- Page 2, lane 26-27: Since it is a general description of F-ATP synthase, it should be "ion-motive force" instead of "proton motive force", since some F-ATP synthases are also Na⁺-driven. The same counts for page 2, lane 42.
- Page 5, lane 123: delete "cryogenic-electron microscopy", since it is defined on page 4, lane 115.
- Page 10, lane 217 "w as" should be "was".
- References: Please correct the following author names:
"Kuhlbrandt" → "Kühlbrandt"
"Gruber" → "Grüber"
"Borsch" → "Börsch"

In summary, the studies described in the manuscript by Sobti et al., are well designed and performed, and contribute to novel insights into the regulatory mechanisms and diversities of such molecular engines.

Reviewer #2

(Remarks to the Author)

This manuscript describes near-atomic-resolution cryo-EM structures of *Pseudomonas aeruginosa* ATP synthase in several rotational states and uncovers two previously unreported features: (i) an unusual ϵ -subunit auto-inhibition mode and (ii) occlusion of the F_o cytoplasmic proton channel by a divalent metal ion. These results are mechanistically and therapeutically important, but two critical issues remain: (1) the absence of a structure for a non-inhibited conformation of the same enzyme and (2) incomplete identification of the bound metal species. I therefore recommend Major Revision.

Major comments

(1) Only the inhibited state is visualised

The study presents the ϵ -clamped, MgADP-inhibited state only. An open (non-inhibited) conformation is essential for drug-design comparisons.

Option 1 – Prepare grids without ADP, perform 3-D classification, and extract any ϵ -released population (atomic resolution is unnecessary).

Option 2 – Determine a cryo-EM map of the ϵ -CTH2-deletion mutant (ϵ^{Δ}) already used in the activity assays.

Option 3 – If neither dataset can be obtained, add an explicit discussion explaining why the non-inhibited state could not be captured and how that limitation affects interpretation.

Even a 4–5 Å map—or a clear rationale for its absence—would substantially strengthen the mechanistic conclusions.

(2) Uncertain assignment of the bound metal ion

Cryo-EM density, ICP-MS, and coordination-site mutants point to Zn²⁺, yet Ni²⁺/Co²⁺/Cu²⁺ cannot be excluded. Please add one of the following:

Option 1 – X-ray anomalous dispersion or XANES/EXAFS at the Zn K-edge.

Option 2 – Cryo-EM-EDS/EELS to detect Zn directly on the grid.

Option 3 – Functional rescue: remove the metal with a Zn-specific chelator (e.g., TPEN), show activity loss, and restore activity by re-adding Zn²⁺ (but not other divalent cations).

Any of these experiments would greatly strengthen the metal assignment. If none can be performed, please (i) state explicitly in the Discussion that the ion cannot be unambiguously confirmed as Zn²⁺ and (ii) delete or revise the sentence in the Abstract that implies Zn²⁺ has been definitively identified.

(3) Statistical reporting

For ATP-synthesis assays and growth curves, give the number of replicates (n), report mean ± SD, and specify the statistical test used (e.g., Student's t-test or one-way ANOVA).

Minor comments (typography / formatting)

p. 12, l. 330 Section heading "DISSCUSSION" → "DISCUSSION".

p. 20, l. 434 "sample incubated at in ice" → "sample incubated on ice".

p. 20, l. 442 "using EPU at of ×59 000" → "using EPU at ×59 000".

p. 21, l. 455 "heterogenous refinements" → "heterogeneous refinements".

With a non-inhibited structure (or a transparent explanation of its absence) and definitive metal identification (or a frank discussion of the remaining uncertainty), the manuscript will make a strong contribution to the field.

Reviewer #3

(Remarks to the Author)

The manuscript by Sobti et al reports the cryoEM structure of the bacterial F1Fo ATP synthase from *P. aeruginosa*. The results are novel and noteworthy on several counts:

1. This is, to my knowledge, the highest-resolution structure of a bacterial ATP synthase published to date. Because F-type ATP synthases play such a key role in molecular bioenergetics and biological energy conversion, insights into the structure

at this level of detail are intrinsically interesting and important.

2. The authors describe several unexpected features of the *P. aeruginosa* ATP synthase, including a previously unknown inhibition site, where the epsilon and gamma subunits interact to stop ATPase activity.

3. Most surprisingly, they find a coordinated metal ion in the cytoplasmic proton channel, which appears to be unique to the pseudomonads. This family includes dangerous human pathogens, and the high-resolution structure of its ATP synthase, which is essential for survival, offers itself as a starting point for drug development.

4. The metal ion was identified as Zn by mass spectrometry, another noteworthy achievement.

The work supports the conclusions and claims (but see below). No additional evidence is needed. The methodology is sound and meets the highest standards in the cryoEM field. The methods are described in sufficient detail for the work to be reproduced.

This is an excellent manuscript that can be accepted for publication essentially as it stands, with the exception of a few passages that would benefit from minor revision:

Line 29: The abstract gives the impression that the overall resolution of the structure is very close to 2 Å. This is certainly the case for the F1 subcomplex, but not for the equally important, but less well investigated and arguably more critical Fo part, where the resolution is 2.4 Å (line 135). In terms of cryoEM structures, this is a large difference. It would therefore be better to give the resolution in the abstract as "2 to 2.4 Å", as is good practice in the field.

Line 54: It is my understanding that the proton does not bind to other c-subunits sequentially but protonates one acidic residue in one c-subunit which then rotates with the ring until it encounters the cytoplasmic half channel where the proton is released to the cytoplasm.

Line 60: The F1 motor is stationary but not immobile. It rocks back and forth around the central stalk (see ref 38). It would be interesting to know if this rocking motion is also observed in the *P. aeruginosa* ATP synthase. The flexible link between the two domains of the delta or OSCP subunit suggests that this motion is a highly conserved feature of all rotary ATP synthases.

Line 216: The ring stoichiometry varies from eight to seventeen (see Schulz et al., Molecular architecture of the N-type ATPase rotor ring from *Burkholderia pseudomallei*. EMBO Rep 18: 526-35 (2017).

Lines 258, 285 and 307: It is true that a bound Zn ion has not been found in this particular position in other ATP synthases. However, a bound metal ion, most likely Zn, is known to bind in a nearby position in the a-subunit of the *Polytomella* ATP synthase (ref. 38). Although the functional role of this bound metal ion is likewise unknown, it would be worth pointing out this most intriguing finding.

Werner Kühlbrandt

Reviewer #4

(Remarks to the Author)

In this manuscript, Sobti et al. have isolated the *Pseudomonas* ATP synthase for structural studies using cryo-EM and identified unique features in the enzyme, which have been validated using biochemical assays. A novel binding site for the C-terminal helices of ϵ subunit and a surprising metal binding site in the cytoplasmic hemi-channel have been resolved. The cryo-EM map qualities are excellent, the manuscript is well written, and the findings are of significant interest to scientists in the Bioenergetics and infectious disease communities. However, I have the following concerns.

Major concerns.

1. Does purified *Pseudomonas* ATP synthase exhibit coupled ATPase activity? Several studies (For example, Sobti et al. 2023, Shah et al. 2013) that report on epsilon inhibition, measure ATPase activity of the purified enzyme with and without the C-terminal region of ϵ . Since this manuscript presents an autoinhibitory mechanism via a distinct epsilon binding site, measuring ATPase activity of WT *Pseudomonas* F1Fo and the $\epsilon 90\Delta$ mutant is important to compare with the *E. coli* enzyme.

2. Lines 423-427 "However, the compound was not observed in the resulting map. Despite this, the addition of compound #5 significantly improved the quality of the cryo-EM grids, facilitating data collection and ultimately enhancing the final map. Therefore, we present the 'apo' enzyme here in the presence of compound #5, as the compound was not detected in this study." – If compound #5 was not detected in the cryo-EM maps (which are of high-enough resolution to detect binding of Zn²⁺), can the authors unambiguously say that addition of compound #5 improved their cryo-EM data quality? I think this statement needs to be reevaluated.

Minor concerns.

1. Line 133 “Fo rotary positions, separated by a $\sim 36^\circ$ rotation of the c-ring (Figure 2, S1 and S2)” – Given that the rotational position of F1 does not change, and the c-ring is homomeric, how was its rotation by $\sim 36^\circ$ measured? It might be a good idea to include this in this information in the Methods section. In addition, how would the authors characterize this substep? Does addition of 10 mM ADP or inhibitor #5 have anything to do with this, or is the energy change for a single c-subunit rotation low enough to occur spontaneously? Some discussion about substep 2 is recommended.
2. Figure 2 – The relative occupancy of the different rotational states should be mentioned in this figure. This information highlights the conformational landscape of rotary enzymes, and I suggest that it be included here (specifically for Fo-substep 2).
3. Figure 4 – Labeling the C-terminal region of alpha/beta and the foot domain of gamma will be helpful for the readers as these are referred to in lines 170-171. Also, which rotational substep/state was used to generate this figure? Is the inhibitory site of epsilon on gamma consistently resolved in all of them?
4. Line 217 – Typo “w as” should probably be was.
5. Figure 7 – Does the cryo-EM density for cD60 suggest a protonated/deprotonated state? A figure panel with a closer look might be helpful in light of the Grothuss mechanism.
6. Figure 8b – Since significant reduction of ATP synthesis is observed in the aE207a mutant, I suggest conducting ICP-MS with this mutant. A significant reduction in the signal for zinc in the mutant would validate the suggested mechanism.
7. Line 315 – “deletion of aAsp288.” Is this meant to be aAsn288?

Version 1:

Reviewer comments:

Reviewer #1

(Remarks to the Author)

The authors have addressed all points in the revised manuscript.

Reviewer #2

(Remarks to the Author)

The authors have addressed my previous concerns with great care and diligence.

In particular:

They have successfully provided new cryo-EM data capturing the non-inhibited state, thereby substantially strengthening the mechanistic conclusions.

They have expanded their ICP-MS analysis and revised the manuscript text to clearly acknowledge the limits of certainty regarding the bound metal ion. This transparent handling of the evidence is commendable.

Statistical reporting has been appropriately improved, and all minor typographical issues have been corrected.

Overall, the manuscript has been significantly improved and revised to a very high standard.

The authors have responded thoroughly and convincingly to all comments, and the revised version presents a clear, well-supported, and impactful study.

I am confident that this work will be of broad interest to the community and makes a strong contribution to the field.

I therefore fully support publication of this manuscript in Nature Communications.

Reviewer #3

(Remarks to the Author)

The authors have answered all my queries and have, over and above, improved the manuscript further by adding new data in response to comments of other reviewers.

I have no further queries or comments and recommend that the manuscript should be accepted for speedy publication.

Werner Kühlbrandt

Reviewer #4

(Remarks to the Author)

My concerns have been adequately addressed by the manuscript's revision and authors' comments. The 11° rotational substep was an interesting revelation in the revised version. I support the publication of this manuscript in Nature Communications.

We thank all reviewers for their helpful and constructive comments and have modified the manuscript along the lines suggested. We have performed further assays and experiments, including: an improved interpretation of the rotational substepping seen in F_o , further ICP-MS, activity assays, and a new cryo-EM dataset in the presence MgATP.

Please find our point-by-point reply to the reviewer comments below. For clarity, reviewers' comments are in black, our replies are in red, and modified text is in *Italic*.

Reviewer #1 (Remarks to the Author):

The *Pseudomonas aeruginosa* F₁F_o-ATP synthase is essential for the non-fermenting pathogen and therefore for the majority of the synthesis of Adenosine triphosphate (ATP). Recent studies by one of the teams have shown, that the c-ring of the F_o-domain can be target by an inhibitor, resulting in ATP depletion of inverted membrane vesicles (IMVs; Fraunfelder et al., 2023). In the presented manuscript by Sobti et al, the authors describe the purification of the *P. aeruginosa* F₁F_o-ATP synthase, the determination of its atomic structure using cryo-EM, design, generation and characterization of the C-terminal truncated ϵ subunit mutant. The study is in line with significant contributions of the Stewart lab to the structural and mechanistic understanding of the *Escherichia coli* F-ATP synthase.

Comments:

- As shown by the Fraunfelder et al., 2023 publication (ACS Inf. Dis.), IMVs of *P. aeruginosa* show moderate ATP-driven proton-pumping activity, indicating ATP hydrolytic activity of the *P. aeruginosa* F₁F_o-ATP synthase. Therefore, an ATP hydrolysis experiment of the purified enzyme in the presence and absence of LDAO would add important information to this manuscript in regard to the activity of the heterologous produced enzyme as well as its ATPase latency and would allow a comparison to the F-ATP synthases from *Mycobacteria* and *Acinetobacter baumannii*.

Understanding ATP hydrolysis in the enzyme is indeed an important aspect that we did not directly address in this study. To provide additional insights, we performed ATP regeneration assays on the purified F₁F_o ATP synthase. We added the following text at the beginning of the Results section: “...shown to be active and *Lauryldimethylamine oxide (LDAO) sensitive in ATP regeneration assays (Figure S1; rate = 7.6 ATP/F₁F_o/sec before and rate = 128.4 ATP/F₁F_o/sec after addition of LDAO)*” and added new supplementary figure to present this data (Figure S1).

- In this context, ATP-driven proton-pumping activity and ATP hydrolysis measurements of the wild-type (WT) and ϵ 90 Δ mutants IMVs would provide more information about the effect of the C-terminus to both activities and the argument that ATPase inhibition is mainly caused by this ϵ CTD.

Again, to address this we performed ATP hydrolysis measurements of the wild-type, ϵ 90 Δ and a subunit mutants using IMVs. The ϵ 90 Δ and ϵ 90 Δ aEC-like mutants showed increase in ATP hydrolysis.

We have now combined these into a functional assays figure (Figure 6), and added to the text: “...and increased ATP hydrolysis (Figure 6c) in inverted membrane

vesicles... ” and “ATP hydrolysis assays on these mutants showed little to no effect, potentially suggesting that the coordination site is involved in synthesis rather than hydrolysis mode. However, as alanine substitutions can be relatively disruptive in their nature, the observed effects may reflect interference with proton translocation in this region rather than direct disruption of ion binding.” and “This mutant derivative construct, designated “aEC-like”, showed ATP synthesis and hydrolysis activity similar to that of single alanine substitution mutants (Figure 6), although this may have been due to lower expression levels (Figure S16). To explore this mutant further and to see if it was linked to ATP hydrolysis, we also combined this aEC-like mutant with the ϵ CTD removed mutant, designated “ ϵ 90 Δ aEC-like”. The mutant showed reduced growth on succinate medium, and the strongest reduction in ATP synthesis and highest increase in ATP hydrolysis (Figure 6). Together, these data suggest that both these regions may be involved in efficient synthesis and hydrolysis of ATP and disrupting them results in dysregulation of the enzyme.”

- The deletion of the ϵ CTD dropped NADH-driven ATP synthesis of the IMVs by about 50%, which is nice. However, considering that the Krebs cycle will contribute to ATP formation, this mutant would still be able to produce about 60% within the pathogen under normal grow conditions, enabling the pathogen to replicate and survive. Figure S10 shows no major effect of this deletion in cell growth under Glucose conditions. Even if a compound would show optimal binding, it would not prevent the bacterium from growth and most importantly would not be bactericidal. This aspect is important for target validation and should be discussed.

- Similarly, the aGlu182Ala, aHis186Ala, aAsp287Ala, aHis289Ala and aGlu207Ala mutants showed some degree of ATP synthesis inhibition compared to the WT IMVs and no major drop in cell growth no matter whether succinate or glucose was present (accept aGlu207Ala under succinate conditions).

The growth curves provided here require careful interpretation. The data reflect the growth of *E. coli* DK8 cells expressing the *P. aeruginosa* ATP synthase, meaning that while *E. coli* can survive with these mutants, the results do not directly inform on *P. aeruginosa* cells. Instead informing more on the function of the enzyme in this setting. *E. coli* DK8 readily ferments glucose even in the absence of ATP synthase, whereas *P. aeruginosa* does not (DOI: 10.1016/S0065-2911(06)52001-6). Our growth curves of *E. coli* DK8 cells expressing the *P. aeruginosa* ATP synthase allow for a positive growth control during the assay, but it also means that the impact of truncation on *P. aeruginosa* cells remains unclear.

Still, in light of this comment we have softened the language in the manuscript and added the following clarification:

Abstract: “Collectively, these findings define novel structural features of *P. aeruginosa* ATP synthase that **could** serve as targets for antimicrobial therapeutics”

Results: “However, such a compound **may not completely prevent bacterial growth and may require combination with other agents.**”

Discussion: “However, their **effectiveness would require validation and compounds based on these regions may be ineffective without combination with other agents.**”

- In this context, it is written in figure legend 7B: “The cytoplasmic proton channel is capped by the C-terminal tail of subunit a, which wraps around a metal”. This channel includes aGlu182Ala, aHis186Ala or aAsp287Ala. Could it be that the

inhibition observed in ATP formation is in part due to the effect in proton transduction and not mainly because of the Zn²⁺-ion?

This indeed could be the case, we have added the following text to the results: *“However, as alanine substitutions can be relatively disruptive in their nature, the observed effects may reflect interference with proton translocation in this region rather than direct disruption of ion binding.”*

• In a cryo-EM structure of *Polytomella* sp. (Murphy et al. (2019) Science) a Zn²⁺-ion near residue aH248 was also resolved, which takes slightly different positions in cryo-EM states P1 and P2. Such data were used in MD simulation studies by Blan and Hummer (PNAS 121 (2024)). These results should be discussed in context to the Zn²⁺-binding and possible role found for the *P. aeruginosa* F1FO-ATP synthase. This Zn²⁺-ion was observed at the other half-channel and therefore was not originally discussed. However, we'd be happy to include this in the manuscript. We have added the following text to the discussion: *“In other cryo-EM and Molecular Dynamics studies on *Polytomella* ATP synthase^{36,55}, a Zn²⁺ cation has been proposed to interact with aHis248 (aGlu230 in *P. aeruginosa*). This site is in the other half-channel (periplasmic) of the F_o-motor and appears to adopt slightly different positions as the c-ring rotates. However, in our cryo-EM study of the *P. aeruginosa* enzyme, we propose that the appearance of the map (Figure 8a) is only consistent with waters in this region. Higher-resolution maps of ATP synthases from various species are now increasingly revealing metal ions bound within key protonation and deprotonation regions. This aspect of ATP synthase function likely requires further study, especially considering that some previous studies have used buffers containing chelating agents, which may have obscured metal binding.”*

• Minor points:

- Page 2, lane 26-27: Since it is a general description of F-ATP synthase, it should be “ion-motive force” instead of “proton motive force”, since some F-ATP synthases are also Na⁺-driven. The same counts for page 2, lane 42.

These and others have been changed to “ion”

- Page 5, lane 123: delete “cryogenic-electron microscopy”, since it is defined on page 4, lane 115.

Deleted

- Page 10, lane 217 “w as” should be “was”.

Fixed

- References: Please correct the following author names:

“Kuhlbrandt” → “Kühlbrandt”

“Gruber” → “Grüber”

“Borsch” → “Börsch”

All these and other errors are fixed now. Sorry, these spellings were taken through PubMed import function.

In summary, the studies described in the manuscript by Sobti et al., are well designed and performed, and contribute to novel insights into the regulatory mechanisms and diversities of such molecular engines.

Reviewer #2 (Remarks to the Author):

This manuscript describes near-atomic-resolution cryo-EM structures of *Pseudomonas aeruginosa* ATP synthase in several rotational states and uncovers two previously unreported features: (i) an unusual ϵ -subunit auto-inhibition mode and (ii) occlusion of the F_o cytoplasmic proton channel by a divalent metal ion. These results are mechanistically and therapeutically important, but two critical issues remain: (1) the absence of a structure for a non-inhibited conformation of the same enzyme and (2) incomplete identification of the bound metal species. I therefore recommend Major Revision.

Major comments

(1) Only the inhibited state is visualised

The study presents the ϵ -clamped, MgADP-inhibited state only. An open (non-inhibited) conformation is essential for drug-design comparisons.

Option 1 – Prepare grids without ADP, perform 3-D classification, and extract any ϵ -released population (atomic resolution is unnecessary).

Option 2 – Determine a cryo-EM map of the ϵ -CTH2-deletion mutant (ϵ^{Δ}) already used in the activity assays.

Option 3 – If neither dataset can be obtained, add an explicit discussion explaining why the non-inhibited state could not be captured and how that limitation affects interpretation.

Even a 4–5 Å map—or a clear rationale for its absence—would substantially strengthen the mechanistic conclusions.

Thank you for these thoughtful suggestions. To capture an open (non-inhibited) conformation, we imaged the same purified F_1F_o ATP synthase after the addition of 10 mM MgATP. This approach has proven successful in our previous studies on the *E. coli* enzyme and was chosen for its high likelihood of yielding comparable results. We have added a new main figure (Figure 5), supplementary figures (Figures S7,8), and corresponding text to describe this finding. Most interestingly, we observed a conformational transition to an “active” ϵ CTD condensed state. We believe this discovery provides valuable insights into the regulation of this enzyme and significantly enhances the manuscript.

The text has been modified as follows:

Abstract: “Lower-resolution maps of the enzyme following incubation with MgATP showed conformational rearrangements of the ϵ subunit during activation.”

*Results: “To further investigate ϵ mediated inhibition in *P. aeruginosa* ATP synthase, we performed cryo-EM on the same purified sample following incubation with 10 mM MgATP instead of 10 mM MgADP (Figures 5, S7 and S8). Although the resulting maps varied in resolution, State 2 yielded reconstructions at ~ 3.5 Å, sufficient to resolve the conformation of each subunit. Consistent with previous studies on the *E. coli* enzyme^{25,26}, we observed a transition of the ϵ CTD from an extended “up” to a condensed “down” conformation, accompanied by a small rotation of the γ subunit. However, unlike the related *E. coli* enzyme (that was only seen in the ϵ CTD “down” state after incubation with 10 mM MgATP), here we captured both ϵ CTD conformations in the same experiment, with the ϵ CTD “up” structure near identical to that presented earlier in this study.”*

Discussion: “Nevertheless, our cryo-EM data showing a transition of the ϵ CTD to a condensed “down” conformation following incubation with MgATP, combined with this functional data, does suggest that it is involved in regulating the enzyme.”

(2) Uncertain assignment of the bound metal ion

Cryo-EM density, ICP-MS, and coordination-site mutants point to Zn^{2+} , yet $Ni^{2+}/Co^{2+}/Cu^{2+}$ cannot be excluded. Please add one of the following:

Option 1 – X-ray anomalous dispersion or XANES/EXAFS at the Zn K-edge.

Option 2 – Cryo-EM-EDS/EELS to detect Zn directly on the grid.

Option 3 – Functional rescue: remove the metal with a Zn-specific chelator (e.g., TPEN), show activity loss, and restore activity by re-adding Zn^{2+} (but not other divalent cations).

Any of these experiments would greatly strengthen the metal assignment. If none can be performed, please (i) state explicitly in the Discussion that the ion cannot be unambiguously confirmed as Zn^{2+} and (ii) delete or revise the sentence in the Abstract that implies Zn^{2+} has been definitively identified.

Our laboratory does not have access to the first two methods proposed. Further, EXAFS is likely impractical due to the high sample concentrations required to generate a strong signal for metal identification, and XANES would not provide significant additional insights beyond what has already been obtained through ICP-MS. Elemental analysis of *P. aeruginosa* (doi: 10.1038/srep13139) show that cobalt is present at very low levels in this organism, with the majority of the metal occurring as the metal chelate cobalamin, making it highly unlikely that cobalt would be a cognate metal ligand. To further support our findings, we have conducted additional ICP-MS analysis, which again suggests zinc as the most likely candidate. Multiple attempts using chelating agents, followed by desalting of the protein complex, and samples analyses by ICP-MS were conducted to increase confidence in elemental composition assessment and assignment of the metal identity. Unfortunately, these approaches proved inconclusive due to issues of residual metal ion contamination in the concentrated samples, while increased washing/dilution of the samples resulted in reduction or loss of the detergent-solubilised protein complex, which in turn limited the precision of the elemental measurements.

Nonetheless, the balance of evidence supports the assignment of a zinc ion at the binding site based on the abundance of first-row metal ions within *P. aeruginosa* and the amino acid composition of the site, which has previously been reported to serve physiological roles in the coordination of zinc and/or manganese

(10.1371/journal.pone.0019510) cations. However, based on this work we cannot definitively identify the ion at this stage and have revised the manuscript text to acknowledge the limits of our certainty. These include:

Abstract: Now reads “Mass spectrometry analyses **suggests** that the physiological metal within the complex is zinc.”

Figure 8 legend: removed “subsequently identified as zinc,”

Results: Added “although our methods cannot assign an identity to this ion unequivocally.”

Discussion: Removed “zinc” from “zinc capped”, and added “In this study, the identity of the ion capping the cytoplasmic proton channel could not be determined unequivocally by ICP-MS or coordination chemistry.”

(3) Statistical reporting

For ATP-synthesis assays and growth curves, give the number of replicates (n),

report mean \pm SD, and specify the statistical test used (e.g., Student's t-test or one-way ANOVA).

As suggested we have now included SD and one-way ANOVA testing. The p values SD and test are listed in the figure legends and Statistical analysis section.

Minor comments (typography / formatting)

p. 12, l. 330 Section heading "DISSCUSION" → "DISCUSSION".

p. 20, l. 434 "sample incubated at in ice" → "sample incubated on ice".

p. 20, l. 442 "using EPU at of ×59 000" → "using EPU at ×59 000".

p. 21, l. 455 "heterogenous refinements" → "heterogeneous refinements".

All these have been corrected now.

With a non-inhibited structure (or a transparent explanation of its absence) and definitive metal identification (or a frank discussion of the remaining uncertainty), the manuscript will make a strong contribution to the field.

We have now included a non-autoinhibited cryo-EM study, extended the ICP-MS analysis, and included a discussion on the uncertainty.

Reviewer #3 (Remarks to the Author):

The manuscript by Sobti et al reports the cryoEM structure of the bacterial F1Fo ATP synthase from *P. aeruginosa*. The results are novel and noteworthy on several counts:

1. This is, to my knowledge, the highest-resolution structure of a bacterial ATP synthase published to date. Because F-type ATP synthases play such a key role in molecular bioenergetics and biological energy conversion, insights into the structure at this level of detail are intrinsically interesting and important.

2. The authors describe several unexpected features of the *P. aeruginosa* ATP synthase, including a previously unknown inhibition site, where the epsilon and gamma subunits interact to stop ATPase activity.

3. Most surprisingly, they find a coordinated metal ion in the cytoplasmic proton channel, which appears to be unique to the pseudomonads. This family includes dangerous human pathogens, and the high-resolution structure of its ATP synthase, which is essential for survival, offers itself as a starting point for drug development.

4. The metal ion was identified as Zn by mass spectrometry, another noteworthy achievement.

The work supports the conclusions and claims (but see below). No additional evidence is needed. The methodology is sound and meets the highest standards in the cryoEM field. The methods are described in sufficient detail for the work to be reproduced.

This is an excellent manuscript that can be accepted for publication essentially as it stands, with the exception of a few passages that would benefit from minor revision:

Many thanks for these highly positive comments.

Line 29: The abstract gives the impression that the overall resolution of the structure is very close to 2 Å. This is certainly the case for the F₁ subcomplex, but not for the equally important, but less well investigated and arguably more critical F_o part, where the resolution is 2.4 Å (line 135). In terms of cryoEM structures, this is a large difference. It would therefore be better to give the resolution in the abstract as "2 to 2.4 Å", as is good practice in the field.

We have changed the abstract to read: *"Here, we present the 2.0-2.4 Å resolution cryo-electron microscopy structures of the ATP synthase from Pseudomonas aeruginosa,..."*

In light of another reviewer's comment, we also now presented the structure after incubation with MgATP and have added the following text which describes the resolution of these structures in the abstract: *"Lower-resolution maps of the enzyme following incubation with MgATP showed conformational rearrangements of the ε subunit during activation"*

Line 54: It is my understanding that the proton does not bind to other c-subunits sequentially but protonates one acidic residue in one c-subunit which then rotates with the ring until it encounters the cytoplasmic half channel where the proton is released to the cytoplasm.

We see how this could have caused confusion, we merely were saying that they bind to c-subunits one after the other. We have simplified the text to: *"Ions drive the rotation of the c-ring by entering a channel that is open only to the periplasm, binding to a conserved acidic residue on a c subunit⁷, which then rotates anticlockwise (when viewed from the F₁-ATPase) until it aligns with a cytoplasmic half-channel, through which the ion is released (Figure 1c)^{8,9}."*

Line 60: The F₁ motor is stationary but not immobile. It rocks back and forth around the central stalk (see ref 38). It would be interesting to know if this rocking motion is also observed in the P. aeruginosa ATP synthase. The flexible link between the two domains of the delta or OSCP subunit suggests that this motion is a highly conserved feature of all rotary ATP synthases.

We did observe this movement, but did not discuss this in the submitted manuscript. We have added the following text to the results, along with a supplementary figure: *"Previous studies have highlighted that rotary ATPases (which F₁F_o ATP synthases are a subclass) are dynamic, with the catalytic "1" motor tilting and rocking relative to the "o" motor during rotation to facilitate smooth coupling³⁴⁻³⁶. Superposition of the P. aeruginosa ATP synthase rotary sub-states resolved here on the F_o motor reveals that similar tilting and rocking of the F₁-ATPase occurs in this organism (Figure S5), with these movements facilitated by twisting and bending of the peripheral stalk."*

Line 216: The ring stoichiometry varies from eight to seventeen (see Schulz et al., Molecular architecture of the N-type ATPase rotor ring from Burkholderia pseudomallei. EMBO Rep 18: 526-35 (2017).

Apologies, we overlooked this finding. The text has been updated.

Lines 258, 285 and 307: It is true that a bound Zn ion has not been found in this particular position in other ATP synthases. However, a bound metal ion, most likely Zn, is known to bind in a nearby position in the a-subunit of the Polytomella ATP

synthase (ref. 38). Although the functional role of this bound metal ion is likewise unknown, it would be worth pointing out this most intriguing finding.

Along with the other reviewer comment, we have now discussed this in more detail: *“In other cryo-EM and Molecular Dynamics studies on *Polytomella* ATP synthase^{36,55}, a Zn²⁺ cation has been proposed to interact with aHis248 (aGlu230 in *P. aeruginosa*). This site is in the other half-channel (periplasmic) of the F_o-motor and appears to adopt slightly different positions as the c-ring rotates. However, in our cryo-EM study of the *P. aeruginosa* enzyme, we propose that the appearance of the map (Figure 8a) is only consistent with waters in this region. Higher-resolution maps of ATP synthases from various species are now increasingly revealing metal ions bound within key protonation and deprotonation regions. This aspect of ATP synthase function likely requires further study, especially considering that some previous studies have used buffers containing chelating agents, which may have obscured metal binding.”*

Werner Kühlbrandt

Reviewer #4 (Remarks to the Author):

In this manuscript, Sobti et al. have isolated the *Pseudomonas* ATP synthase for structural studies using cryo-EM and identified unique features in the enzyme, which have been validated using biochemical assays. A novel binding site for the C-terminal helices of ϵ subunit and a surprising metal binding site in the cytoplasmic hemi-channel have been resolved. The cryo-EM map qualities are excellent, the manuscript is well written, and the findings are of significant interest to scientists in the Bioenergetics and infectious disease communities. However, I have the following concerns.

Major concerns.

1. Does purified *Pseudomonas* ATP synthase exhibit coupled ATPase activity? Several studies (For example, Sobti et al. 2023, Shah et al. 2013) that report on epsilon inhibition, measure ATPase activity of the purified enzyme with and without the C-terminal region of ϵ . Since this manuscript presents an autoinhibitory mechanism via a distinct epsilon binding site, measuring ATPase activity of WT *Pseudomonas* F1Fo and the $\epsilon 90\Delta$ mutant is important to compare with the *E. coli* enzyme.

As suggested by another reviewer, we have now performed ATP hydrolysis assays on the $\epsilon 90\Delta$ mutant (Figures 6c).

2. Lines 423-427 “However, the compound was not observed in the resulting map. Despite this, the addition of compound #5 significantly improved the quality of the cryo-EM grids, facilitating data collection and ultimately enhancing the final map. Therefore, we present the 'apo' enzyme here in the presence of compound #5, as the compound was not detected in this study.” – If compound #5 was not detected in the cryo-EM maps (which are of high-enough resolution to detect binding of Zn²⁺), can the authors unambiguously say that addition of compound #5 improved their

cryo-EM data quality? I think this statement needs to be reevaluated.

An initial aim of this study was to determine the binding mode of compound #5 using cryo-EM. Consequently, during early sample preparation screening, we included compound #5, solubilized in DMSO. Interestingly, its addition significantly improved particle distribution and contrast in the cryo-EM images. While we also solved the structure without compound #5 or DMSO in the presence of 10 mM MgADP, those maps were of lower quality (~3 Å resolution for State 1) but did contain all the key features presented in this study.

As requested by another reviewer, we also collected cryo-EM data after adding 10 mM MgATP. Though with this dataset we omitted compound #5 and DMSO to provide as a text to address this comment. Some particles remained in the autoinhibited state, and these maps again revealed the same position of the ϵ CTD and the capped cytoplasmic proton channel, corroborating the primary findings of this manuscript.

Our laboratory does not focus on method development, so we have not pursued the relationship between compound #5 or DMSO and improved map quality systematically. The enhanced map/grid quality we observed could be attributed to compound #5, DMSO, a combination of the two, or potentially an unidentified contaminant from the compound's synthesis or purification. While compound #5 itself was not observed in the maps (possibly due to interactions with detergent or the grid support surface) this was not the primary focus of our study, and we have avoided further speculation. We believe the methods section provides sufficient context for this observation.

Minor concerns.

1. Line 133 "Fo rotary positions, separated by a ~36° rotation of the c-ring (Figure 2, S1 and S2)" – Given that the rotational position of F1 does not change, and the c-ring is homomeric, how was its rotation by ~36° measured? It might be a good idea to include this in this information in the Methods section. In addition, how would the authors characterize this substep? Does addition of 10 mM ADP or inhibitor #5 have anything to do with this, or is the energy change for a single c-subunit rotation low enough to occur spontaneously? Some discussion about substep 2 is recommended.

This rotation angle was initially misstated on our part. We originally reported an angle of ~36° as it appeared to our eyes to correspond to the approximate rotation of one c subunit. However, after reanalysing the data carefully, we have now determined the rotation angle to be ~11°, which is highly noteworthy and we believe greatly improves the findings of the manuscript. We thank the reviewer for this comment.

To calculate this angle accurately, we aligned the maps and models on the a subunit and applied the methods now detailed in the manuscript and Figure S16. The calculated rotation angle was 10.8° (which we quote as ~11° in the main text), and the substeps show distinct interactions with the a subunit. In Fo substep 1, a c subunit carboxylate (cAsp60) is aligned with the cytoplasmic channel, while in Fo substep 2, a different c subunit carboxylate is aligned with the periplasmic channel (Figure S4). This indicates that one substep facilitates cytoplasmic proton release,

and the other supports periplasmic proton uptake, effectively separating these events during rotation.

Further analysis of cAsp60 rotamers around the c-ring (also suggested by this reviewer below), revealed that the side chain can either point toward or away from the centre of the ring depending on its position (Figure S17).

These observations align well with findings from prior single molecule rotation (DOI: 10.7554/eLife.70016) and crystallographic studies (DOI: 10.1038/nsmb.2284). We have summarised these findings in a new figure (Figure 9) and added the following text to the manuscript.

Abstract: “Focused classification of the F_o motor resolves distinct $\sim 11^\circ$ sub-steps in the c-ring, corresponding to protonation and deprotonation events.”

*Results: “As well as defining the cytoplasmic and periplasmic half-channels, the cryo-EM maps also highlight a sub rotation of the F_o -motor. In State 2 (Figure 2), focused classification methods produced two maps of the F_o -motor in which the c-ring had rotated relative to the a subunit (Figure 9 and S4). Analysis of this rotation showed a rotation of $\sim 11^\circ$, as calculated by superposing on the a subunit (Figure S16). Notably, in one position, the proton-carrying carboxylate group on the c subunit (cAsp60) was located adjacent to the periplasmic channel, whereas in the other position, it is adjacent to the cytoplasmic channel (Figure 9 and S14). This arrangement suggested that the enzyme pauses in each of these positions, potentially protonating at one and deprotonating at the other. Most interestingly, this rotation and proposed arrangement is near identical to that observed and hypothesized in single molecule rotation assays on *E. coli* F_1F_o ATP synthase, where a proton translocation dependent 11° c-ring rotation steps have been observed⁴⁶. The density for carboxylates in cryo-EM maps is often poor, and even though the resolution of these maps was relatively high, we could not assign unequivocally the rotamer position of the cAsp60 in all cases. However, the features of the maps suggested that the aspartate adopts different rotamers depending on its protonation state and whether it is exposed to lipids or proteins, either pointing toward or away from the c-ring (Figure S17), consistent with observations from crystallographic studies at varying pH⁴⁷.”*

Discussion: “Additionally, these maps support a Grotthuss mechanism for proton movement through the periplasmic proton channel, and F_o stepping with separated protonation and deprotonation events.” and “Additionally, we have visualized directly a water lined periplasmic half-channel consistent with a Grotthuss proton-transfer mechanism and captured discrete F_o substeps that separate protonation and deprotonation.”

2. Figure 2 – The relative occupancy of the different rotational states should be mentioned in this figure. This information highlights the conformational landscape of rotary enzymes, and I suggest that it be included here (specifically for F_o -substep 2).

We have included the relative number of particles in each state in this figure. We have chosen not to comment further on these observations here. While we believe they are interesting and meaningful, previous attempts by us to interpret similar findings have been met with criticism from reviewers.

3. Figure 4 – Labeling the C-terminal region of alpha/beta and the foot domain of

gamma will be helpful for the readers as these are referred to in lines 170-171. Also, which rotational substep/state was used to generate this figure? Is the inhibitory site of epsilon on gamma consistently resolved in all of them?

We have now labelled the regions suggested and added the following text to the legend: “(*F₁* rotary state 1 shown here)”.

The inhibitory site is consistent across all MgADP maps as well as the autoinhibited maps observed after addition of MgATP.

4. Line 217 – Typo “w as” should probably be was.

This has been corrected

5. Figure 7 – Does the cryo-EM density for cD60 suggest a protonated/deprotonated state? A figure panel with a closer look might be helpful in light of the Grothuss mechanism.

This has been addressed in our response to Minor Point 1.

6. Figure 8b – Since significant reduction of ATP synthesis is observed in the aE207a mutant, I suggest conducting ICP-MS with this mutant. A significant reduction in the signal for zinc in the mutant would validate the suggested mechanism.

While we agree with the reviewer’s suggestion, due to the volume of protein complex required for comprehensive ICP-MS analyses, this approach is not feasible at this time. Fully resolving the precise function of this region will require significant improvements in protein yield to permit further investigation. Nonetheless, our present work highlights its importance and provides preliminary insights. In response to another reviewer’s feedback, we have revised the manuscript to use more cautious language regarding the identity of the ion.

7. Line 315 – “deletion of aAsp288.” Is this meant to be aAsn288?

Sorry for the confusion here, this was indeed meant to be aAsn288 and this has been corrected in the text.